# Indecision and recency-weighted evidence integration in non-clinical and clinical settings

**Magdalena del Río** [1,2] ✉, **Nadescha Trudel**[1,2], **Gita Prabhu** [2], **Laurence T. Hunt**[3], **Michael Moutoussis** [2], **Raymond J. Dolan** [1,2,4,7] & **Tobias U. Hauser** [1,2,5,6,7] ✉

Biases in information gathering are common in the general population and reach pathological extremes in paralysing indecisiveness, as in obsessive–compulsive disorder (OCD). Here we adopt a new perspective on information gathering and demonstrate an information integration bias whereby there is over-weighting of most recent information via evidence strength updates (ΔES). In a crowd-sourced sample ($N$ = 5,237), we find that a reduced ΔES weighting drives indecisiveness along an obsessive–compulsive spectrum. We replicate this attenuated ΔES weighting in a second lab-based study ($N$ = 105) that includes a transdiagnostic obsessive–compulsive spectrum encompassing OCD and generalized anxiety patients. Using magnetoencephalography (MEG), we trace ΔES signals to a late neural signal peaking at ~920 ms. Critically, highly obsessive–compulsive participants, across diagnoses, show an attenuated neural ΔES signal in mediofrontal areas, while other decision-relevant processes remain intact. Our findings establish biased information weighting as a driver of information gathering, where attenuated ΔES is linked to indecisiveness across an obsessive–compulsive spectrum.

Knowing when to decide is difficult[1,2]. From selecting a movie on a streaming platform to buying a car, whenever there is uncertainty about the 'right' choice, gathering more information usually helps us make better decisions. But being indecisive and spending too much time gathering information is also disadvantageous and carries substantial opportunity and energy costs[3].

In humans, information gathering biases are common, often expressed as gathering too little or too much information[4,5]. Dramatically inflated information-gathering biases are a feature of psychopathology and believed to drive core symptoms[4,6–11]. For example,

indecisiveness and persistent doubt can be debilitating aspects of obsessive–compulsive disorder (OCD)[12,13]. Excessive deliberation also extends beyond core OCD symptoms and can impact everyday wellbeing and function[14].

While indecisiveness in OCD is traditionally assessed using clinical interviews[15], recent literature has provided a more quantitative approach based on using information gathering tasks[4,10,16–23]. Studies using the latter have shown that patients with OCD, as well as non-clinical participants with high obsessive–compulsive (OC) scores, manifest elevated levels of information gathering before committing

[1]Max Planck UCL Centre for Computational Psychiatry and Ageing Research, University College London, London, UK. [2]Wellcome Centre for Human Neuroimaging, University College London, London, UK. [3]Department of Experimental Psychology, University of Oxford, Oxford, UK. [4]State Key Laboratory of Cognitive Neuroscience and Learning, IDG/McGovern Institute for Brain Research, Beijing Normal University, Beijing, China. [5]Department of Psychiatry and Psychotherapy, Faculty of Medicine, University of Tübingen, Tübingen, Germany. [6]German Centre for Mental Health (DZPG), Tübingen, Germany. [7]These authors jointly supervised this work: Raymond J. Dolan, Tobias U. Hauser. ✉e-mail: magdalena_del_rio@brown.edu; t.hauser@ucl.ac.uk

to a decision. While similar effects have been observed in naturalistic settings[23], the precise neurocognitive mechanisms underlying the expression of indecisiveness remain unknown.

A relevant field of study, largely disconnected from information gathering, is that of evidence accumulation in decision making[24]. Here it is considered that evidence for different options increases gradually with incoming information, and a decision is made when evidence for an option hits a decision threshold[24–29]. The latter is believed to collapse with time, meaning that decisions become increasingly liberal, which is often framed in terms of an urgency signal[30–33]. These effects are observed in a range of information-gathering contexts despite striking differences in tasks and timescales[34–36]. Importantly, recent studies suggest that evidence does not accumulate linearly as a function of decision-relevant information, but in a recency-biased manner such that the most recent evidence is over-weighted[37–40]. Determining whether such biases also exist in information gathering and whether they act as drivers for indecisiveness is an important question.

In this study, we investigate the neurocognitive contributors to information gathering across two samples and tasks which we previously collected in the lab. To pre-empt the results, we find that, in addition to urgency-like signals, evidence is integrated non-monotonically, with what we term as evidence strength updates ($\Delta$ES) critically determining when a participant decides to decide. Furthermore, across both studies, we find that participants along an OC spectrum rely less on these $\Delta$ES both in clinical and non-clinical samples. Using magnetoencephalography (MEG), we show that this attenuation of $\Delta$ES is mirrored in the brain, with neural $\Delta$ES representations in mediofrontal areas being less evident in OC whereas other critical contributors to information gathering remain intact. We conclude that indecisiveness along an OC spectrum is driven by a reduced reliance on most recent information.

## Results

### Cognitive contributors to information gathering

To first ascertain a link between information gathering and OC symptoms in the general population, we conducted a crowd-sourced study ($N$ = 5,237 at the time of analysis). To this end, we implemented an information-gathering task on an app for handheld electronic devices (www.brainexplorer.net), where participants had to decide which of two stimuli was more plentiful across 25 hidden locations (Fig. 1a). Participants were free to sample as many of the locations as they wanted before committing to a decision, such that we could use the number of draws before a decision as an index of indecisiveness.

First, to better understand the cognitive contributors to participants' information-gathering behaviour, we used general linear mixed models (GLMMs) to predict on each draw whether a participant would continue sampling information or make a decision ($p$(decide)) using a series of cognitive and individual differences predictors (see Methods for the full model). In this task, information accumulates non-monotonically, allowing us to distinguish the contribution of different cognitive factors (for example, current information, total evidence, number of draws) to commitment to a decision. To test whether a recency bias[41–43] is present in an information-gathering context, we distinguished total evidence strength on the previous draw ($ES_{d-1}$; absolute cumulative evidence difference across draws [1 to d−1] for the current majority) from the most recent change in evidence strength or evidence strength update ($\Delta$ES; Fig. 1a). This is analogous to a distinction between previous expectation and prediction error, that is, the mismatch between the previous expectation and the actual observation, which is central to frameworks and models on processes spanning from perception to learning across domains[44,45].

Model comparison favoured the model with $\Delta$ES and $ES_{d-1}$ as separate predictors over a model where all evidence was aggregated in $ES_d$ as a single predictor (difference in Akaike information criterion ($\Delta$AIC) = 1.709 × 10^6; difference in Bayesian information criterion ($\Delta$BIC) = 1.709 × 10^6), supporting the idea of a (recency-) weighted information integration. We found that both the current update in signal strength ($\Delta$ES, $\beta$ = 1.105, $P$ < 0.001, 95% confidence interval (CI) 1.086–1.123; Fig. 1c) and previous evidence strength ($ES_{d-1}$; $\beta$ = 1.042, $P$ < 0.001, 95% CI 1.029–1.055) positively predicted whether participants would commit to a decision, but that the former predicted it more strongly. A separate GLMM predicting $p$(decide) from the current evidence at draw d, d−1 and d−2 relative to the chosen option showed a decreasing contribution of current evidence for lagged draws ($ES_d$: $\beta$ = 0.648, $P$ < 0.001, 95% CI 0.635–0.662; $ES_{d-1}$: $\beta$ = 0.253, $P$ < 0.001, 95% CI 0.245–0.262; $ES_{d-2}$: $\beta$ = 0.195, $P$ < 0.001, 95% CI 0.185–0.204; Supplementary Fig. 1a). This accords with a recency bias in information gathering, analogous to that found across other decision-making domains[37,38,44,46]. It is particularly striking as this task was designed so that all information remained present on the screen irrespective of when it was gathered, such that this recency bias is not accounted for by working memory limitations or related explanations.

In keeping with the idea of collapsing decision boundaries or urgency signals[4,30,47,48], we also included the draw number as a predictor. Draw number additionally predicted whether participants would make a decision ($\beta$ = 1.789, $P$ < 0.001, 95% CI 1.752–1.826), such that participants were more likely to stop gathering information the longer they had already spent gathering information (irrespective of how strong the evidence is) and, in addition, if they (recently) received stronger evidence favouring one option over the other.

### OC-linked indecisiveness is due to attenuation of evidence weighting

Next, we investigated whether an OC spectrum was linked to indecisiveness. Taking the number of draws before making a decision as an indicator of indecisiveness[4,34,36,49,50], we found, as hypothesized, that participants with higher OC symptoms (as determined by the Obsessive–Compulsive Inventory-Revised (OCI-R) total score[51]) gathered more evidence before making a decision (Fig. 1b; $\rho_s$ = 0.040, $P$ = 0.004, 95% CI 0.012–0.067), while not showing any difference in accuracy for these decisions ($\rho_s$ = −9.998 × 10^{-4}, $P$ = 0.942, 95% CI −0.028 to 0.026, $BF_{01}$ = 38.81, robustness region RR > 0.036). Thus, in this information-gathering dataset, we find that indecisiveness relates to OC symptoms across the general population.

Next, we investigated which of the above identified cognitive contributors explained differences in task behaviour in high-OC participants. We found that OC symptoms were linked to reduced weighting of previous evidence $ES_{d-1}$ ($ES_{d-1}$ × OCI-R: $\beta$ = −0.053, $P$ < 0.001, 95% CI −0.067 to −0.040) and particularly of $\Delta$ES ($\Delta$ES × OCI-R: $\beta$ = −0.085, $P$ < 0.001, 95% CI −0.104 to −0.067). There was no significant main effect of OC symptoms on sampling decisions ($p$(decide); Fig. 1d; $\beta$ = 0.033, $P$ = 0.265, 95% CI −0.025 to 0.092, $\Delta$BIC = 2,205 comparing GLMM with and without main effect of OC), which would imply a simple tendency not to decide; and, contrary to our initial assumptions[4,52], there was no significant effect on urgency (as captured by $N_{draws}$ × OCI-R: $\beta$ = −0.016, $P$ = 0.406, 95% CI −0.053 to 0.021, $\Delta$BIC = 1,786 comparing GLMM with and without $N_{draws}$ × OCI-R interaction effect). We found similar but weaker effects of OC symptoms on evidence weighting when comparing a subset of participants with a self-reported OCD diagnosis with those who explicitly declared no lifetime psychiatric disorder (see Supplementary Information 'Self-reported OCD diagnosis effects on information gathering in smartphone population sample'). To follow up on the recency effect suggested by these results, we targeted this by fitting a separate GLMM predicting $p$(decide) from the current evidence relative to the chosen option at draw d, d−1 and d−2 per participant (see Methods). We found that a difference in the beta weight for the evidence at draw d and the beta weight for the evidence at draw d−1 correlated negatively with the OCI-R score ($\rho_s$ = −0.069, $P$ < 0.001, 95% CI −0.099 to −0.038), supporting the notion of a

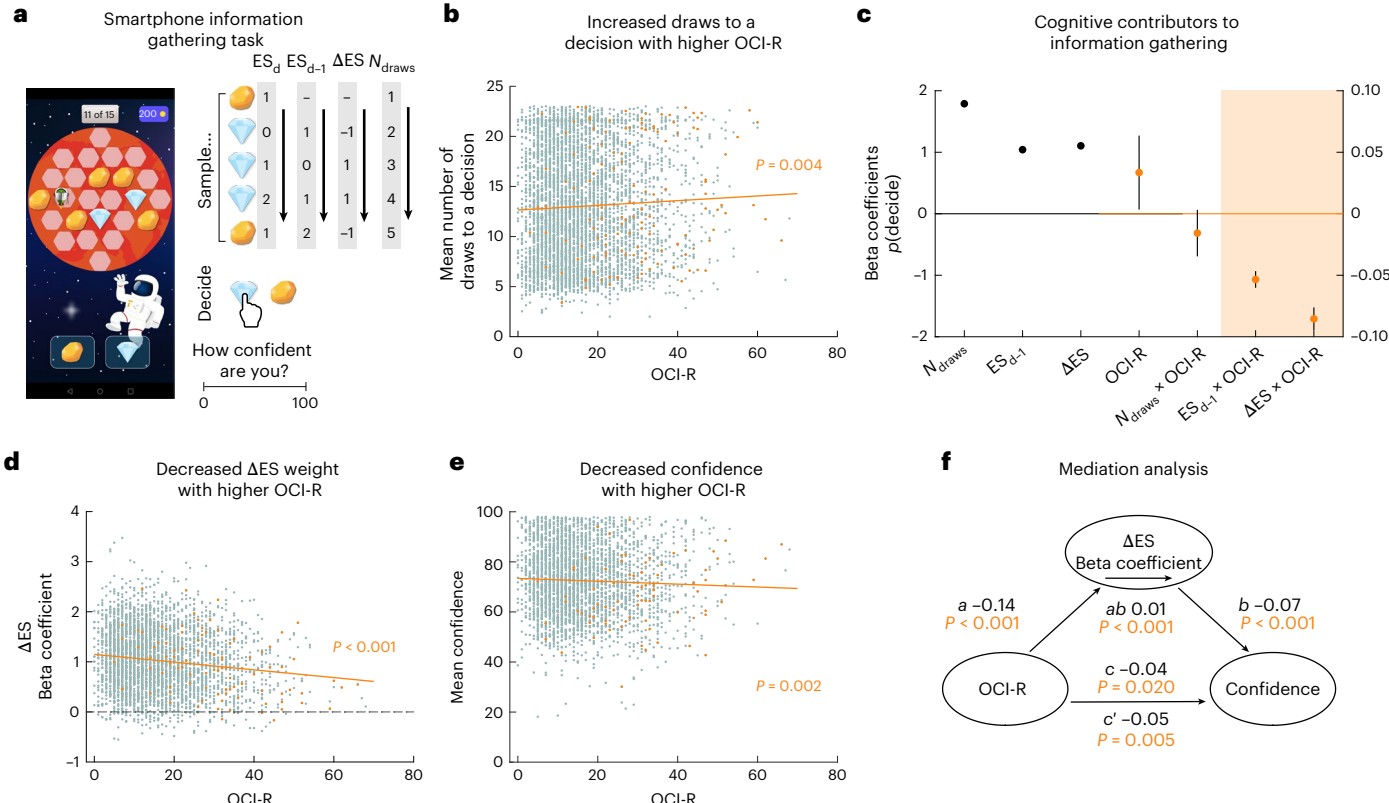

**Fig. 1 | OC symptoms are linked to increased information gathering in a smartphone population study. a**, The participants' task was to determine which of two possible gems was more abundant (screenshot of the task on the Brain Explorer app, https://brainexplorer.net/). They were free to uncover as much evidence as they wished, by tapping on locations on the grid, until they decided to commit to a binary choice between one of two gems. An example sequence of samples and corresponding evidence accumulation process for cumulative evidence strength at the previous draw $ES_{d-1}$ and the evidence strength update $\Delta ES$ is depicted on the right. Here evidence strength at draw d $ES_d$ is quantified as the cumulative difference in evidence for the two gems with respect to the gem that is more abundant at draw d (for example, here 3 diamonds minus 2 yellow gems constitutes an evidence strength of 1 at draw 5). Cumulative evidence strength at the previous draw $ES_{d-1}$ is defined as the lagged ES by one draw, and evidence strength update $\Delta ES$ is quantified as the signed difference between the cumulative ES at draw d−1 and draw d. Participants rated their confidence after every binary choice, then received 100 points for correct responses or lost 100 points for incorrect responses. **b**, Within a sample of the general population ($N = 5,237$), individuals with higher OC symptoms sampled more information before committing to a decision ($\rho_s = 0.040$, $P = 0.004$). Data points pertaining to individuals with a self-reported OCD diagnosis are shown in orange. **c**, The probability of making a decision p(decide) was predicted from experimental factors as well as individual differences in OC symptoms using a general linear mixed model (GLMM, $N = 5,237$; data are presented as beta coefficients ± standard error (s.e.), scale adjusted to show main effects of experimental factors on the left and effects associated with individual differences in OC on

the right y axis). The experimental factors (left y axis) comprised current draw number, cumulative evidence from the start of the game until the previous draw $ES_{d-1}$, and the evidence strength update $\Delta ES$. More draws and higher $ES_{d-1}$ and $\Delta ES$ increased the probability of making a decision p(decide) (all $P < 0.001$). Individual differences in obsessive–compulsive symptoms (right y axis) were quantified as the main effect of the OCI-R score and the interactions of the OCI-R score with the experimental factors. The analysis showed that OC symptoms per se did not have a significant effect on p(decide) ($P = 0.265$), and that people with high OC symptoms did not weight the current draw number differently in deciding to commit to a choice ($P = 0.406$); instead, they weighted both previous and current evidence less ($P < 0.001$, highlighted by shaded area). The latter effect is illustrated in **d**. **d**, To obtain estimates of sensitivity to evidence at a single-participant level, we fit a general linear model (GLM) predicting p(decide) separately for each participant. Convergent with the significant interaction effect in the GLMM, those with higher OC symptoms had reduced beta coefficients for $\Delta ES$, that is, they weighed $\Delta ES$ less in deciding to commit ($\rho_s = -0.121$, $P < 0.001$, $N = 3,901$ after removing outliers, defined as any participant with any beta values 1.5× interquartile range (IQR) above the third quartile or below the first quartile). **e**, Individuals with higher OC symptoms also rated their confidence in their decisions as being lower on average ($\rho_s = -0.054$, $P = 0.002$, $N = 3,293$, see exclusion criteria in Methods). **f**, A mediation analysis suggests that sensitivity to $\Delta ES$ partially mediates the association between OC symptoms (OCI-R total score) and confidence ratings, such that lower confidence is partially explained by altered evidence integration ($N = 3,293$). All tests are two-sided and not corrected for multiple comparisons.

decreased recency effect. This means that high-OC participants were no more likely to decide per se, or express an altered urgency signal, but instead showed an attenuated impact of the most recent information sample. In other words, weaker evidence strength updates seem to be driving indecisiveness along the OC spectrum.

**Lowered confidence in high-OC participants mediated by attenuated $\Delta ES$**

Having identified key drivers of indecisiveness, we next assessed whether this was a purely behavioural effect or whether it also affected participants' perception of their performance or insight. To assess

this, we asked participants to rate their post-choice confidence (after committing to a decision, but before viewing the outcome; Fig. 1a). We found that OC symptoms were also linked to confidence, with high-OC participants having lower confidence (Fig. 1e; $\rho_s = -0.054$, $P = 0.002$, 95% CI −0.088 to −0.020, $N = 3,293$, after excluding additional participants based on confidence data criteria, see Methods). This means that the OC spectrum was not only linked to indecisive information gathering but also to a lack of confidence, a finding that aligns with evidence of impoverished confidence in other domains[53]. It also chimes with the notion that indecisiveness is closely linked to subjective doubt.

Lastly, using mediation analysis (https://github.com/canlab/MediationToolbox (refs. 54,55)), we evaluated the link between the reduced ΔES weight from our information-gathering analysis and confidence findings. We found that the ΔES weighting (using beta estimates from non-hierarchical models; Fig. 1d) was associated with task confidence ($b = -0.066$, $Z = -3.721$, $P < 0.001$, 95% CI $-0.100$ to $-0.031$; Fig. 1f). Specifically, we found that ΔES partially mediated the influence of OC symptoms on confidence ($ab = 0.010$, $Z = 3.775$, $P < 0.001$, 95% CI $0.005-0.015$; $c = -0.043$, $Z = -2.335$, $P = 0.020$, 95% CI $-0.079$ to $-0.008$; $c' = -0.052$, $Z = -2.809$, $P = 0.005$, 95% CI $-0.088$ to $-0.016$; Fig. 1f). This means that the lower confidence in high-OC participants is, at least in part, influenced by the same underlying process as that of attenuated ΔES.

## Cognitive contributors to information gathering replicate in a lab-based sample

As the above dataset was primarily derived from a self-selected population of non-clinical participants, we next asked whether the findings also held true in a more targeted patient sample. Thus, we analysed a previously collected independent sample that included patients assessed in the lab ($N = 105$), using a novel adaptation of the above information-gathering paradigm (Fig. 2a).

We first replicated the findings related to the cognitive contributors to information gathering per se (Fig. 2c). In a GLMM similar to the one above (see Methods for the equation), we found that both ΔES ($\beta = 0.830$, $P < 0.001$, 95% CI $0.791-0.869$) and previous evidence strength ($\beta = 0.516$, $P < 0.001$, 95% CI $0.475-0.557$) positively predicted when participants would make a decision. Likewise, we replicated the effect of current draw number ($\beta = 2.367$, $P < 0.001$, 95% CI $2.232-2.503$), supporting the notion of an emerging decision urgency. A noteworthy extension here was that this lab version featured two different finite horizons, meaning that each game would end after a probabilistically predetermined number of draws (either 4–8 in the 'short' or 10–14 in the 'long' horizon condition). We found that this horizon condition significantly predicted participants' decision-making ($\beta = 1.602$, $P < 0.001$, 95% CI $1.457-1.747$). In other words, participants are not only more likely to commit to a decision if they (recently) received more evidence for one of the options and if they had already gathered information for longer, but also if they are incentivized to trade off speed for accuracy.

## Attenuated ΔES is also linked to OC symptoms in a transdiagnostic clinical sample

Next, we investigated how these predictors link to an OC spectrum in a sample consisting of a clinical group of OCD patients ($N = 29$) and generalized anxiety disorder (GAD) patients ($N = 17$), as well as non-patient cohorts with varying degrees of OC symptoms (high OC: $N = 20$, low OC: $N = 20$, controls: $N = 19$). This recruitment strategy allowed for a broad sampling of OC symptoms not only in OCD patients but also in related disorders as well as non-clinical high-OC participants. A similar approach has previously revealed striking dissociations between transdiagnostic dimensions and traditional diagnostic categories[56]. To derive a transdiagnostic and dimensional OC symptom measure, we factor analysed key psychiatric questionnaires collected across the entire sample, yielding a 3-factor solution where one factor primarily reflected OCD and related symptoms (Methods and Supplementary Fig. 3). On average, OCD patients ranked highest on this factor, followed by high-compulsive non-patients and GAD patients, while low-compulsive non-patients and controls ranked the lowest (Fig. 2b).

Investigating how this transdiagnostic OC factor linked to cognitive contributors of information gathering, we again replicated the effect of ΔES (ΔES × OC: $\beta = -0.049$, $P = 0.013$, 95% CI $-0.088$ to $-0.010$; Fig. 2c,d), such that high-OC participants weighted the update in evidence strength less in their decision to declare for a stimulus. Similar, albeit less robust, effects were seen when comparing OCD patients to matched controls alone ($\beta = -0.102$, $P = 0.046$, one-sided test, 95% CI

$-0.221$ to $0.017$). In addition, by fitting a separate GLMM predicting $p$(decide) from the evidence at draw d, d−1 and d−2 relative to the chosen option per participant, we again found a decreasing contribution of current evidence for lagged draws ($ES_d$: $\beta = 0.531$, $P < 0.001$, 95% CI $0.500-0.562$; $ES_{d-1}$: $\beta = 0.232$, $P < 0.001$, 95% CI $0.205-0.259$; $ES_{d-2}$: $\beta = 0.246$, $P < 0.001$, 95% CI $0.224-0.267$; Supplementary Fig. 1b), and that the difference in the beta weight for the evidence at draw d and the beta weight for the evidence at draw d−1 correlated negatively with OC symptoms ($\rho_s = -0.187$, $P = 0.034$, one-sided test, 95% CI $-0.395$ to $0.042$).

As was the case for the previous population sample, we find no significant main effect of OC symptoms ($\beta = 0.055$, $P = 0.318$, 95% CI $-0.053$ to $0.162$, ΔBIC = 57.641, comparing the GLMMs with and without the OC main effect), or any significant interaction with the current number of draws ($N_{draws} \times OC$: $\beta = -0.098$, $P = 0.155$, 95% CI $-0.234$ to $0.037$, ΔBIC = 90.411, comparing the GLMMs with and without any 2- or 3-way $N_{draws} \times OC$ interaction effects) or with the horizon condition (Horizon × OC: $\beta = -0.074$, $P = 0.320$, 95% CI $-0.219$ to $0.072$, ΔBIC = 233.312, comparing the GLMMs with and without any 2- or 3-way horizon × OC interaction effects). In this sample, the interaction of OC symptoms with previous evidence was not significant ($ES_{d-1} \times OC$: $\beta = -0.023$, $P = 0.272$, 95% CI $-0.064$ to $0.018$, ΔBIC = 203.803, comparing the GLMMs with and without any 2- or 3-way $ES_{d-1} \times OC$ interaction effects), in contrast to the first smartphone sample. In sum, this further supports the notion that an over-reliance on most recent information during information gathering is attenuated in high-OC participants, including patients.

## Attenuated ΔES is primarily linked to obsessions in OCD patients

Lastly, we asked whether an attenuated weighting of ΔES was linked to symptom severity, especially in patients with OCD. To this end, we correlated the scores from the clinical Y-BOCS interview (only conducted in OCD patients by trained researchers) with the ΔES effect on information gathering. Interestingly, we found that attenuated weighting of ΔES was significantly associated with symptom severity for obsessions ($\rho_s = -0.475$, $P = 0.012$, 95% CI $-0.696$ to $-0.166$, Bayes factor for the alternative hypothesis $BF_{10} = 0.101$, robustness region RR > 0.138), but not for compulsions ($\rho_s = -0.205$, $P = 0.306$, 95% CI $-0.548$ to $0.198$, Bayes factor for the null hypothesis $BF_{01} = 5.457$, RR > 0.268; total score trended at $\rho_s = -0.342$, $P = 0.081$, 95% CI $-0.616$ to $-0.016$, $BF_{10} = 0.133$, RR > 0.187). This difference in effect size suggests that the attenuated weighting of recent information may be primarily linked to an obsession dimension, in line with the idea that underweighting of information about the unlikely nature of the obsessions promotes their perpetuation.

To assess how generalizable this effect was, we went back to the complete samples for both in-person and online datasets, and found that ΔES attenuation related to obsessions from the OCI-R questionnaire both in the smartphone population sample ($\rho_s = -0.083$, $P < 0.001$, 95% CI $-0.1144$ to $-0.0535$) as well as in the lab-based sample (including patients and non-patients: $\rho_s = -0.226$, $P = 0.030$, 95% CI $-0.416$ to $-0.013$). However, these effects were less specific (Supplementary Fig. 4), possibly due to range effects across the wide spectrum of patients and non-patients.

Thus, our findings indicate that attenuated ΔES is not only associated with OC symptoms across a wide OC spectrum but also linked to symptom severity of obsessions within OCD patients.

## The brain processes decision-relevant factors sequentially

To better understand the brain processes relevant to information gathering, we concurrently acquired MEG data from all participants in our in-person study. Here, the validity of our behavioural model leads to the prediction that we could decode (5-fold cross-validated iterative lasso regression with whole-brain MEG[57], see Methods) each of the relevant cognitive factors used to predict information gathering, as described

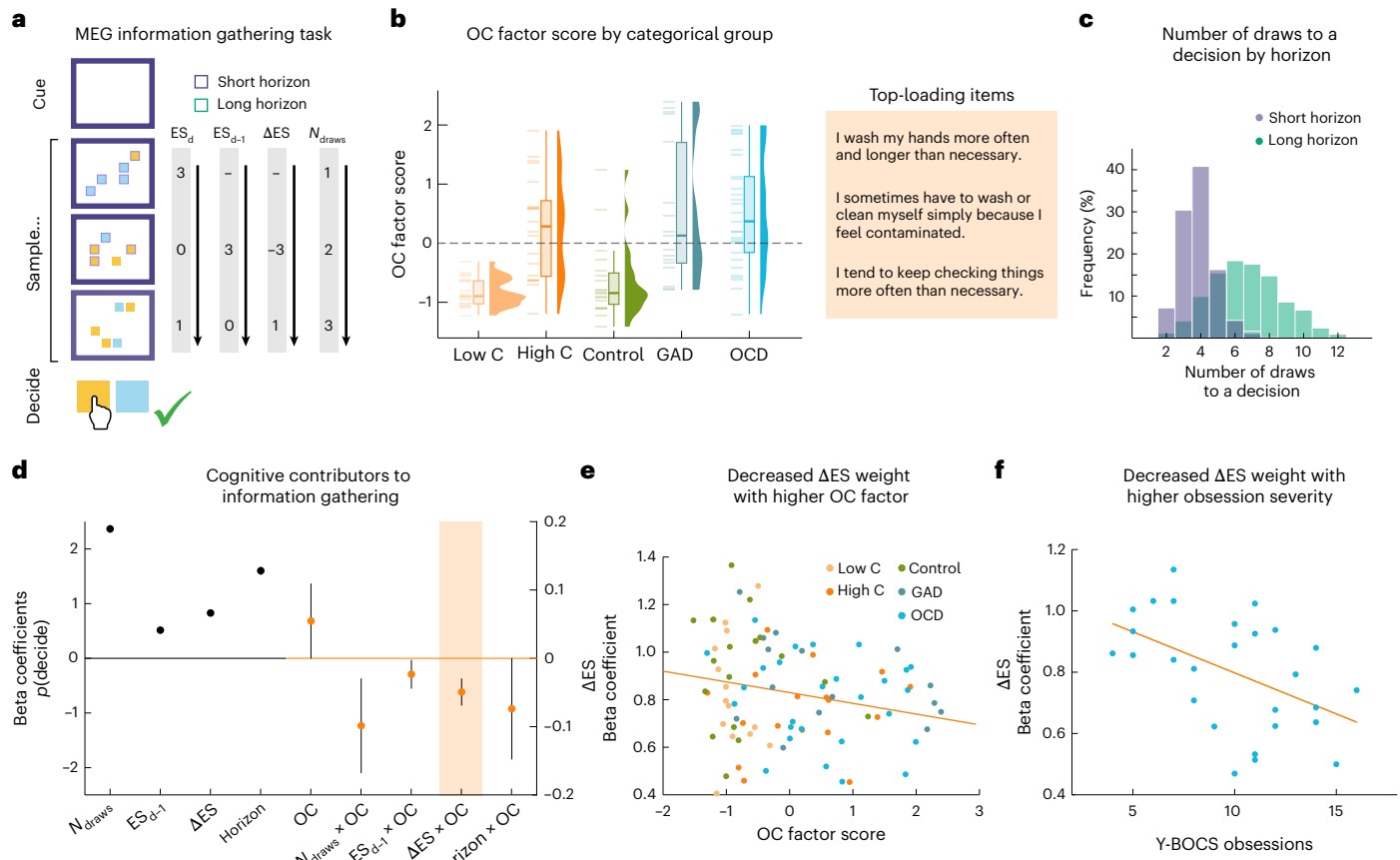

**Fig. 2 | Reduced ΔES weighting replicated in transdiagnostic dimensional patient–control sample. a**, Participants' goal was to determine which of two possible stimuli was more abundant. They were presented with a sequence of draws until they decided to declare for a stimulus. Each draw consisted of five stimuli at a time in varying proportions, allowing for the sequences to be constructed so as to reduce the collinearity between the number of draws, current and previous evidence, which in turn enables us to study their specific representations in the brain. Here again, evidence strength (ES) at draw d is quantified as the cumulative difference in evidence for the 2 gems with respect to the gem that is more abundant at draw d (for example, here 4 blue gems minus 1 yellow gem constitutes an evidence strength of 3 in the first draw, and 8 yellow gems minus 7 blue gems yields an evidence strength of 1 in the third draw). Cumulative evidence strength in the previous draw $ES_{d-1}$ is defined as the lagged ES by one draw, and evidence strength update ΔES is quantified as the signed difference between the cumulative ES at draw d−1 and draw d. The maximum number of draws varied between 4 and 8 in the short-horizon condition and between 10 and 14 in the long-horizon condition. The horizon condition was cued by the colour of the frame (different colours were presented in the task). Participants received 2 points for a correct response, lost 2 points for an incorrect response, and lost 1 point for non-decisions. Points were later translated into bonus payments. **b**, To pool across clinical and non-clinical groups, we reduced the dimensionality of 7 questionnaires using an exploratory factor analysis ($N = 105$). This revealed a 3-factor solution, where the second factor mapped primarily onto items in questionnaires assessing OCD (OCI-R and Padua Inventory-Washington State University Revision (PI-WSUR)). We used this OC factor to quantify individual differences across all participants, plotted here by categorical group, that is, for participants from the general population with low and high obsessive–compulsive scores ('low C' and 'high C', respectively), healthy controls ('Control'), participants with generalized anxiety disorder ('GAD') and

participants with OCD ('OCD'). Boxplots show the median (centre line) and IQR; whiskers extend to the largest value within 1.5× IQR. **c**, The number of draws to a decision varied across trials and participants in the short- and long-horizon condition ($N = 105$). Note that the number of draws to a decision per se was not associated with the transdiagnostic OC factor ($\rho_s = -0.117$, $P = 0.236$), yet this is not surprising given the highly constraining horizon manipulation. **d**, The probability to make a decision was predicted in a GLMM from experimental factors as well as variability in the OC factor ($N = 105$, data are presented as beta coefficients ± s.e., scale adjusted to show main effects of experimental factors on the left and effects associated with individual differences in OC on the right $y$ axis). Experimental factors (left $y$ axis) comprised the number of draws, the cumulative evidence strength from the start of the game until the previous draw $ES_{d-1}$, ΔES and the horizon condition (controlling for interactions of the horizon condition with all other experimental factors). More draws, higher previous and current evidence strength, and a short-horizon condition all increase the probability to make a decision (all $P < 0.001$). Individual differences in the OC factor (right $y$ axis) are quantified as the main effect of the OC factor score and its interaction with the experimental factors. The GLMM shows no significant effect of the OC factor per se ($P = 0.318$), or differences in weighting the current number of draws ($P = 0.155$), the horizon condition ($P = 0.320$), or the previous evidence ($P = 0.272$) in making a decision. Conversely, it shows that those with higher OC factor scores weight ΔES less in making a decision ($P = 0.013$, highlighted by shaded area). **e**, Illustration of the association between ΔES weighting and the OC factor (using individual GLMs per participant; $N = 92$ after outlier exclusions were applied to all beta coefficients in the GLM, defined as any participant with any beta values 1.5× IQR above the third quartile or below the first quartile). **f**, OCD patients with stronger obsession symptom severity had more attenuated weighting of ΔES ($\rho_s = -0.475$, $P = 0.012$, $N = 27$). All tests are two-sided and not corrected for multiple comparisons.

above, namely: ΔES, total evidence strength at draw d−1 $ES_{d-1}$, horizon and number of draws.

We found evidence for a representation of each of the aforementioned factors in our MEG data (Fig. 3a; $P < 0.05$, cluster-based

permutation tests at $t > 3.1$), supporting the idea that the brain tracks these variables. Interestingly, these representations were instantiated at different times within a trial following a sequential order, where pre-existing latent factors were represented first, and novel

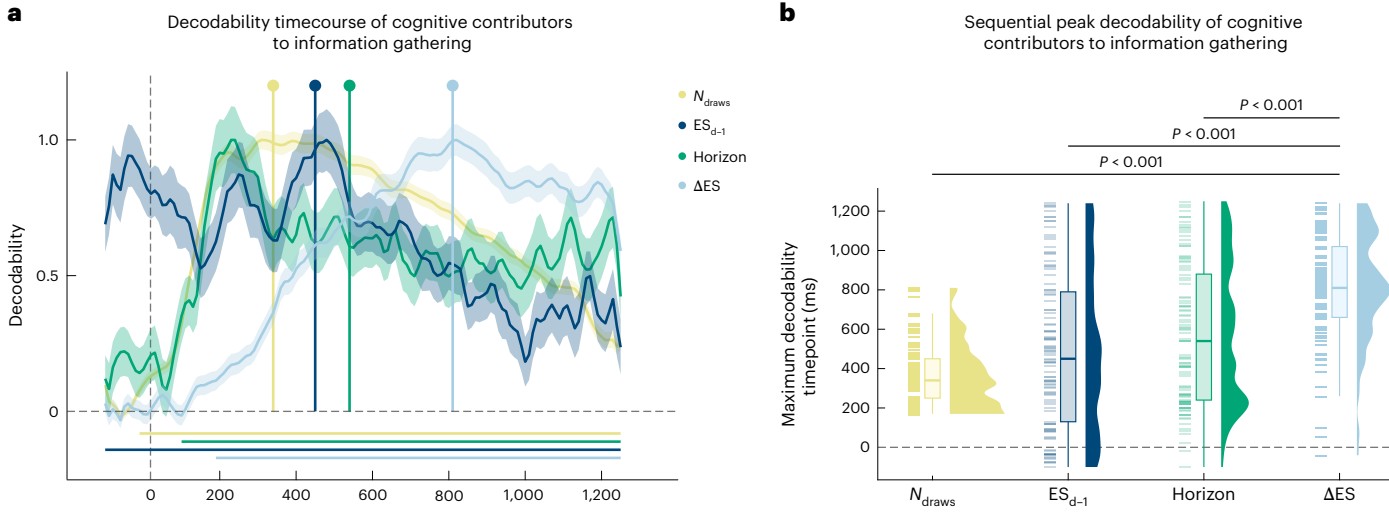

**Fig. 3 | Sequential decodability of cognitive contributors to information gathering from whole-brain MEG activity. a**, Cognitive factors are predicted from MEG activity using lasso regression following an iterative method described previously[57] ($N = 105$). The decoded variables of current number of draws, horizon condition, cumulative evidence strength from the start of the game until the previous draw $ES_{d-1}$ and $\Delta ES$ were all significantly decodable ($P < 0.05$, two-sided cluster-based permutation test at threshold $t > 3.1$; denoted by horizontal lines). Peak decodability timepoints, quantified as the group average of individual maximum values, are indicated by lollipops. Correlation values were normalized by dividing these by the maximum correlation value for visualization purposes. Note that all statistical tests were performed before normalization. Shaded areas represent the standard error. **b**, Maximum decodability is achieved earlier in the trial for factors independent of the currently presented stimulus (current number of draws, horizon condition, the cumulative evidence from the start of the game until the previous draw) than for those dependent on current evidence processing ($\Delta ES$, all $P < 0.001$, $N = 105$). Boxplots show the median (centre line) and IQR; whiskers extend to the largest value within 1.5× IQR. Pairwise $t$-tests are one-sided and not corrected for multiple comparisons.

information that needed to be computed on the basis of these latent representations was represented later in a trial. Specifically, we found that the representations of the number of draws, horizon and the total evidence at draw d−1 preceded the representation of $\Delta ES$ (comparison of maximum decodability timepoint; number of draws vs $\Delta ES$: $t(104) = -14.773$, $P < 0.001$, Cohen's $d = -1.983$, 95% CI −2.311 to −1.651; horizon vs $\Delta ES$: $t(104) = -5.387$, $P < 0.001$, Cohen's $d = -0.777$, 95% CI −1.056 to −0.497; $ES_{d-1}$ vs $\Delta ES$: $t(104) = -6.568$, $P < 0.001$, Cohen's $d = -0.947$, 95% CI −1.231 to −0.662; one-sided tests; Fig. 3b). Importantly, the representation of the currently shown evidence (quantified as the relative evidence for the two alternative response options in the current draw) also preceded the representation of $\Delta ES$ ($t(104) = -8.711$, $P < 0.001$, Cohen's $d = -1.202$, 95% CI −1.494 to −0.907; one-sided test; Supplementary Fig. 5c), as would be expected when considering that $\Delta ES$ is computed on the basis of both previous total evidence and current evidence. In addition, we observed that latent variables that are not dependent on currently presented information manifest as a more sustained representation (for example, horizon), consistent with the fact that they do not need to be computed on the fly after each draw. Thus, our findings demonstrate a cascade of neural information processing and representation that aligns with the evolution of cognitive processes we deciphered from participants' behaviour.

### Reduced neural signatures of ΔES in high-OC participants mirror behavioural findings

Having established the decodability of cognitive contributors to information gathering, we next tested whether these representations were altered in high-OC participants. More concretely, given a reduced effect of $\Delta ES$ on behaviour in the latter group, we hypothesized a reduced strength of this representation in the brain. On this basis, we assessed how transdiagnostic OC symptoms relate to the decodability of $\Delta ES$ by predicting decodability at each timepoint from the three factor scores in individual GLMs. Using cluster-based permutation testing, we found that $\Delta ES$ representations were attenuated in high-OC participants, whereby two time periods reached significance, namely,

420–470 ms and 530–560 ms post stimulus onset (Fig. 4a,b; $P < 0.025$ for a two-sided test, cluster-based permutation tests at $t > 2.1$). Neither decodability of the variable nor individual differences in its decodability appear to be driven exclusively by processes related to the draw wherein participants actually declare, as opposed to the longer-term evidence accumulation process, as both decodability and individual differences associated with the OC factor remained consistent after omitting the data for the draw immediately preceding the response (Supplementary Fig. 6a). Likewise, the results held for the cumulative evidence strength for the current majority $ES_d$, as opposed to the update in evidence strength $\Delta ES$ (Supplementary Fig. 6b). When analysing other variables, we did not find any links with the OC spectrum. This means that, based on behavioural findings, the $\Delta ES$ representation is attenuated in participants with high OC symptoms.

### Attenuation of ΔES representations arises from mediofrontal areas

Lastly, we investigated the likely regional origins of an attenuated OC spectrum-related $\Delta ES$ signal. While we found a relatively widespread activation pattern for $\Delta ES$ per se (Fig. 4c, left panel), the attenuation in high-OC participants was pronounced in mediofrontal areas (Fig. 4c, right panel; see Supplementary Fig. 7 for the results for the second time window and Supplementary Figs. 8 and 9 for alternative methods). This suggests that the observed indecisiveness may arise from altered $\Delta ES$ processing in mediofrontal areas.

## Discussion

Indecisiveness and doubt are common experiences in everyday life but can reach debilitating extremes in mental health conditions, notably OCD. Some aspects of these symptoms can be quantitatively captured as excessive information gathering, yet their roots remain unclear[4,10,16,18,19,21–23]. Here, using a crowd-sourced behavioural dataset in combination with data from a novel neuroimaging task, we show that attenuated weighting of most recent information by means of evidence strength updates ($\Delta ES$) drives altered information gathering

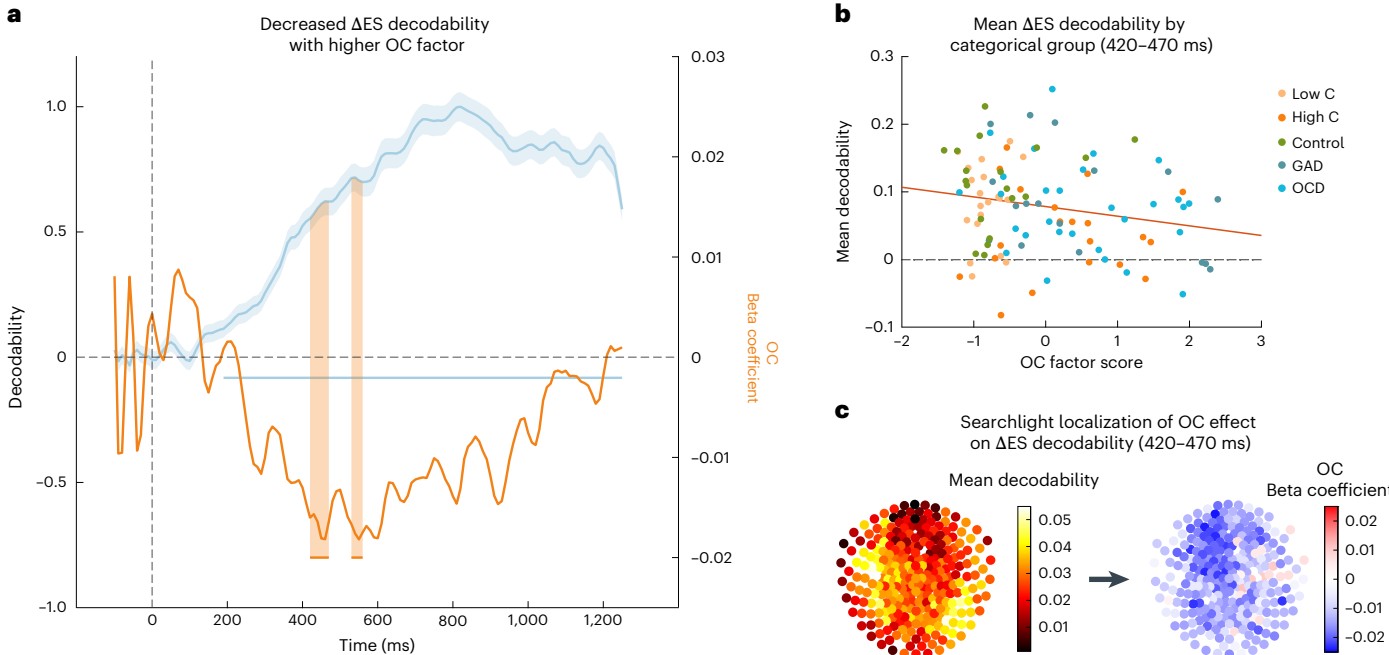

**Fig. 4 | Reduced decodability of ΔES from MEG activity linked to OC symptoms.** **a**, ΔES can be decoded from MEG activity using an iterative multivariate lasso regression (left *y* axis: decodability, measured as the correlation between trained and tested data at each timepoint in the epoch, plotted in blue as group-level mean decodability, with shaded area representing the standard error of the mean, *N* = 105; horizontal lines denote significant time clusters (*P* < 0.05 in a two-sided cluster-based permutation test at threshold *t* > 3.1)). Individual differences in OC factor predict significantly decreased decodability within 420–470 ms and 530–560 ms post stimulus (right *y* axis: beta coefficient for the OC factor predicting the decodability of ΔES at each timepoint in the epoch from the 3 factor scores in a GLM per timepoint, plotted in orange, *N* = 105; horizontal lines denote significant time clusters (*P* < 0.05 in a two-sided cluster-based permutation test at threshold *t* > 2.1)). **b**, Illustration of the attenuated ΔES decodability within 420–470 ms (*N* = 105), plotted here by categorical group, that is, for participants from the general population with low and high obsessive–compulsive scores ('low C' and 'high C', respectively), healthy controls, participants with GAD and participants with OCD. Note that this is merely a visualization of the significant difference found in **a**, showing individual data points. **c**, To investigate the spatial distribution of the sensors contributing to the decoding of ΔES within the time windows where we find individual differences (420–470 ms post stimulus onset), we used searchlight analyses (*N* = 105). ΔES are represented in a wider network comprising frontal, mediocentral and occipital areas (left). The attenuation of ΔES representations in high-OC participants was primarily driven by mediofrontal sensors, supporting the idea of altered ΔES processing in mediofrontal areas in high-OC participants (right).

across an OC spectrum. Our finding of attenuated ΔES was evident at the level of both behaviour and brain, and was also evident dimensionally and transdiagnostically across both patients and those with non-clinical symptoms.

Excessive information gathering has been extensively investigated in relation to OCD and OC-like symptoms[4,10,16,18,19,21–23], yet the precise mechanisms remain unclear. Much previous work on information gathering (including our own) has focused on altered decision criteria, such as collapsing decision thresholds or urgency signals[4,34], ignoring the possibility that evidence integration (that is, how information gathered at different timepoints is combined) might be implicated. We show that information integration shows a strong recency effect, where most recent information (by means of ΔES) exerts a much bigger impact on deciding, and when to decide, than more remote information. A recency weighting has been argued to confer an advantage under certain signal strength conditions[58] or in dynamic contexts where the latest information is most representative[59]. It is also well aligned with similar recency biases found across domains that span perception, memory and mood[37,38,44,46]. Interestingly, the nonlinear accrual of evidence has been widely accepted in the reward learning domain[60,61], and the present results indicate that similar mechanisms are also at play during information gathering. Crucially, ΔES is also represented in the brain, as a late electrophysiological component peaking at ~920 ms after new information is revealed. Importantly, this ΔES representation peaks following the representation of other decision variables from which it is computed (specifically, newly revealed information and previous evidence), further supporting a putative sequential cascade of information processing[62,63].

A mediofrontal localization of ΔES decodability is strongly reminiscent of that found in previous studies on information sampling during value-based decisions, where consistency of current evidence with a participant's belief is signalled by anterior cingulate cortex (ACC) activation in single-cell recordings from monkeys[64] and in an equivalent human magnetic resonance imaging paradigm[65]. In line with the strong effect of ΔES on the probability of committing to a decision, ACC signals peak immediately before the final decision in monkeys[64]. While the signal we show here does not depend crucially on the brain activity immediately before commitment to a decision (Supplementary Fig. 6a), it should be noted that this possibility would be best addressed using a response-locked analysis, as opposed to the stimulus-locked approach we chose, and remains a question for future studies. More broadly, the localization of the ΔES decodability effect is in line with accumulating evidence for alterations in fronto-striatal circuitry in OCD, whereby these findings are often related to heightened error monitoring and altered reward prediction error responses[48,66,67].

The ΔES measure may also fit within a broader belief-updating or predictive coding framework, provided the evidence strength for the current majority maps onto a participant's current belief. If this assumption holds, ΔES can be cast as a type of prediction error (PE), in that it captures the deviation from the expectation formed in the previous draw. While different PEs have been described beyond the classic reward prediction errors, from perceptual to cognitive[44,45], ΔES is different as it is not based on explicit feedback for a completed action but rather on novel information available during the decision-making process. As such, its relationship to a traditional conceptualization of

PEs needs to be established. This also applies to its implementation in the brain—while classic reward PEs have been associated with dopamine signalling, the extent to which dopamine influences information gathering is less clear[34,68,69], and other candidate neurotransmitter systems, such as noradrenaline or serotonin, may conceivably contribute to the formation of ΔES[34,36].

Critically for this study, we find attenuation of ΔES in people with (clinically and non-clinically) high OC symptoms and we propose that it is the likely driver underlying excess information-gathering behaviour found by draws to a decision[4,10,18–23] (although others have reported fewer draws to a decision in OCD[16,17]). Importantly, we find a closer association between the ΔES weighting and the OC dimension, than with draws to a decision alone, supporting the notion that this is the driving factor, while the latter behavioural marker is probably also under the influence of many other cognitive contributors, such as stringent constraints, urgency signals and clear speed–accuracy trade-offs (as found in our MEG experiment). The finding of attenuated ΔES in OC, without a significant effect on draws to a decision, in the more constrained MEG experiment (cf. horizon conditions) also speaks to this, and accords with previous literature showing intact adjustments in speed-accuracy trade-offs in OCD patients[4,22]. Our finding of an attenuated neural representation of ΔES in high-OC participants also suggests that it may provide a sensitive biomarker. Indeed, the mediofrontal regions which appear most implicated in the attenuation of the ΔES representation, provide a tantalizing target for more focused future investigations, as well as for potential therapeutic interventions (for example, using electrical or magnetic stimulation[70]).

A diametrically opposite information-gathering bias prominent in mental health disorders is jumping to conclusions (JTC), particularly seen in patients with psychosis[9,71–73]. Whether this JTC bias is driven by similar or distinct neurocognitive mechanisms remains unexplored. It is interesting, however, that similar imbalanced weighting hypotheses have been proposed for JTC[74–76], along with the notion of altered PE processing in both psychosis and schizophrenia[77].

In this study, we focus on transdiagnostic OC effects within the general population as well as across different clinical diagnoses. While many of our (primarily behavioural) effects hold in more traditional patient vs controls subsamples, these tended to be less strong (for example, MEG differences did not reach significance, data not shown), possibly due to a much smaller sample size. Conversely, we find that OC symptoms are not exclusive to OCD patients but are also prominent in some GAD patients and non-clinical, high-OC participants in our sample (Fig. 2b). While data on indecisiveness in clinically diagnosed GAD is, to the best of our knowledge, lacking, our findings align with previous reports of high indecisiveness in undiagnosed individuals with probable GAD[78] as well as in those with high levels of symptoms of anxiety and depression[79,80]. Thus, we propose that indecisiveness and excessive information gathering may represent transdiagnostic symptoms not limited to OCD.

Nevertheless, it is yet to be determined whether these information-gathering biases are best described using transdiagnostic approaches, as adopted in previous studies where dimensional measures outperformed traditional diagnostic categorical comparisons[56]. Specifically, it remains to be established, first, whether our findings hold in traditional OCD patient case–control studies, and second, to what extent these hold across diagnoses. Indecisiveness can also be a symptom of both depression and dependent personality disorder[12], yet understanding an across-diagnosis overlap in symptom phenomenology and underlying mechanisms will require future studies that include these populations alongside those with OCD and GAD.

Further, indecisiveness manifests in different ways and with different degrees of severity, and our highly abstract task does not capture this complexity. The strength of this abstraction is that it enables the isolation of fundamental evidence accumulation processes and meets neuroimaging constraints, yet it comes at the cost of ecological

generalizability. While previous work has linked increased real-life information gathering to OC symptoms, and the current paradigm shows robust and replicable effects in relation to OC symptoms, it will be important to establish how well our findings translate to real-life settings. Nevertheless, we previously found that information gathering in a version of this task without any sampling constraints is linked to indecisiveness (as rated in the clinician-rated Y-BOCS[52]). At the opposite end of the spectrum, JTCs in a similar paradigm are consistently linked to clinically reported delusions, indicating good ecological validity for this symptom. However, a more targeted ecological validation of the paradigm could help refine our understanding of the throughline from our findings to differences in real-life information gathering in OCD.

In sum, within a crowd-sourced sample as well as a lab-based sample including clinical patients, we provide evidence of an attenuated weighting of ΔES, both at the behavioural and neural level, linked to transdiagnostic OC symptoms. This novel neurocognitive marker highlights the potential contribution of altered evidence integration to excessive information gathering and indecisiveness across health and mental illnesses, potentially opening avenues to improve behavioural and brain-based interventions for these pervasive symptoms.

## Methods

We collected two types of data: a non-clinical sample completed a gamified information-gathering task using the Brain Explorer app for handheld devices as well as the OCI-R. A second in-person sample including OCD and GAD patients together with matched controls and low- and high-compulsive non-patients completed an analogous information-gathering task while their MEG activity was recorded. In both online and in-person studies, human participants completed variants of information-gathering tasks[81] in which they were asked to indicate which of two possible stimuli was more plentiful. They had the choice to either continue sampling additional information or stop and make a binary choice between the two stimuli. Classically, the key metric in this task is 'draws to a decision', that is, the number of draws or samples the participant chooses to view before deciding for either stimulus. We experimentally varied the amount of current and past information and the maximum allowed draws to a decision to assess their impact on information gathering and the relationship with OC symptoms.

### Participants

**Smartphone population sample.** At the time of analysis, 8,670 participants completed the gamified information-gathering task on Brain Explorer (www.brainexplorer.net), of whom 6,743 had also completed the OCI-R (data collected between November 2020 and August 2023). We analysed each participant's first complete game (N = 15 trials) that fulfilled the inclusion criteria, which comprised a mean number of draws between 2 and 23 over the 15-trial game, accuracy in the binary choice between the two stimuli above 80%, and a minimum of 3 unique values in the total set of 15 numbers of draws to a decision, resulting in a final sample size of 5,237.

Additional exclusion criteria were applied to conduct a further analysis on participants' confidence ratings in their binary choice between the two possible stimuli, which they provided immediately after each decision, on a slider ranging from 0 to 100. These criteria comprised a mean confidence rating over the 15-trial game between 15 and 98, and a minimum of 3 unique values in the total set of 15 confidence ratings, yielding a sample size of 4,302. Here, participants with any outlier beta coefficients in the GLM (see below), defined as values 1.5× interquartile range (IQR) above the third quartile or below the first quartile, were removed, resulting in a final sample size of 3,293.

**In-lab MEG sample.** A total of 115 participants completed the MEG study (data collected between April 2015 and August 2018). OCD and GAD patients were recruited through NHS services, charities and

advertisements. To ensure comparable backgrounds, controls were recruited in areas of similar socioeconomic status to the patients'. Exclusion criteria were current use of antipsychotic medication, severe learning disability and comorbidities including current psychosis or bipolar disorder, substance abuse disorder and personality disorders other than obsessive–compulsive personality disorder, as well as self-disclosed autism spectrum, tic or Tourette disorder. As part of the recruitment process through NHS clinics, clinicians pre-screened potential participants on the basis of the inclusion criteria. Those considered eligible were subsequently screened by a study staff member, focusing on existing diagnoses and MEG inclusion criteria. All participants underwent an extensive clinical interview, conducted by experienced researchers at UCL, comprising a structured clinical interview (SCID)[82], the Y-BOCS interview[15] (which assesses clinician-rated symptom severity, only for OCD patients) and 7 questionnaires, namely: the Barratt Impulsiveness Scale (BIS[83]), Beck Depression Inventory II (BDI-II[84]), Intolerance of Uncertainty Scale (IUS[85]), OCI-R[51], PI-WSUR[86], State and Trait Anxiety Inventory (STAI[87]) and Frost Multidimensional Perfectionism Scale (FMPS[88]). The final decision to include a participant was based on a discussion of the results of the interview among the research group members. All participants with OCD and GAD fulfilled the corresponding diagnostic criteria for only one of these disorders[12].

Data from the high- and low-OC groups have been reported previously[89]. Briefly, participants were recruited from a population-based sample of young people (U-CHANGE study; www.nspn.org.uk)[90,91] based on the revised Padua Inventory–Washington State University Revision (PI-WSUR[86]).

Two participants were excluded because they did not meet the clinical criteria for GAD, one participant was excluded because of comorbid OCD and GAD, one was excluded due to fully missing questionnaire data. One additional participant was excluded due to an excessive number of non-decision trials (~55%, compared to a mean of ~20% across the remaining participants), resulting in below-chance performance. Five further participants were excluded due to technical difficulties and/or poor MEG data quality. A total of 105 participants were thus included in the analysis (see Supplementary Fig. 2 for a visualization of the exclusion process). The final groups comprised OCD patients ($N = 29$), GAD patients ($N = 17$), matched controls ($N = 19$), and non-clinical high-compulsive ($N = 20$) and low-compulsive groups ($N = 20$). Group demographics, medication percentages and comorbidities for the final sample included in the analysis are reported in Supplementary Table 1.

The study was approved by the Research Ethics Committees at University College London (UCL; protocol number 16711/002) and the NHS (protocol number 15/LO/1361), and all participants provided written informed consent.

## Task
Participants in both online and in-person studies completed an information-gathering task where their goal was to determine which of two possible stimuli was more plentiful. Each draw consisted of a new sample of the stimuli, after which the participant made the decision to either continue sampling additional information or stop and make a binary choice between the two stimuli.

**Smartphone population sample.** During each game (Fig. 1a), participants viewed a grid of 25 locations, each of which could contain one of two possible stimuli, here gems of different shape and colour. They could view which of two possible gems was hidden at a given position by tapping on it on the grid, or make a binary choice indicating which of the two gems was more plentiful by tapping on the corresponding gem at the bottom of the screen. Uncovered gems remained on the screen until the participant made a binary choice. There were no constraints or penalties on the response time or the number of draws. After each decision, participants gave a confidence rating using a slider and then

received feedback on their performance in the form of a reward or a loss. They won 100 points for every correct choice and lost 100 points for every incorrect choice. Each session consisted of interactive instructions, which included a simplified mock game, followed by the 15 games that were included in the analysis.

**In-lab MEG sample.** To better disentangle cognitive contributing factors and their neural representations, we designed an entirely novel task optimized for MEG. In this version, the two possible products were always blue zircons or gold nuggets, represented respectively by blue and yellow squares on a grey background. Evidence samples consisted of 5 squares at a time which varied in their proportion of yellow to blue, and were presented automatically every 1,250 ms, that is, participants did not actively gather information as in the Brain Explorer task. Participants were instructed that samples were recovered by a robot which may run out of batteries, such that here there was a variable finite number of draws per game, termed the horizon. The maximum number of draws varied between 4 and 8 in the short-horizon condition, and between 10 and 14 in the long-horizon condition. These conditions were cued by a different coloured frame (green or pink, assigned randomly for each participant). There were 80 games in each horizon condition, randomly intermixed. Once participants committed to a choice, they made their response by pressing one of two buttons on a MEG-compatible button box. The length of the draw sequence was presented in full independently of when the participant made their decision; in other words, if the participant made a decision before the last draw, the draw sequence continued until it reached its predetermined end. Participants received thorough instructions and completed 12 games as part of a practice block that included the horizon manipulation before beginning the experiment, such that they acquired a good intuition about the horizon length already during the practice block. They received 2 points for a correct response, lost 2 points for an incorrect response, and lost 1 point for non-decisions. They could track their performance and corresponding rewards by means of a blue bar at the bottom of the screen, which started at the 50% mark. They were told that each time the bar reached the maximum threshold of 100%, they would receive an additional £0.50 at the end of the experiment.

## OC symptoms assessment
**Smartphone population sample.** Participants completed the OCI-R[51] within the Brain Explorer app.

**In-lab MEG sample.** Participants completed seven self-report questionnaires assessing psychiatric symptoms: BIS[83], BDI-II[84], IUS[85], OCI-R[51], PI-WSUR[86], STAI[87] and FMPS[88]. Participants additionally completed the matrix and vocabulary subtests of the Wechsler Abbreviated Scale of Intelligence (WASI[92]), and Edinburgh inventory[93] data were collected to assess handedness. Missing responses in the questionnaires were imputed with the mean response, provided the number of missing items did not exceed 20% of the items in the corresponding scale.

We conducted an exploratory factor analysis (EFA) on the 210 individual items that compose the seven psychiatric symptom questionnaires to reduce the dimensionality of the data. The EFA solution was estimated using maximum likelihood and oblimin rotation based on the heterogeneous correlation matrix to account for both continuous and binary correlations using the psych package in R[94]. The number of factors was determined on the basis of the Cattell–Nelson–Gorsuch index[95], that is, the eigenvalue at which the greatest difference between two successive slopes occurs, visible as a sharp elbow in a scree plot, implemented with the nFactors package in R[96]. Although the sample size resulted in a relatively low ratio of observations to items, adequate sample size for factor analyses ultimately depends on the correlation structure in the data. Indeed, in this case the Kaiser–Meyer–Olkin (KMO) Measure of Sampling Adequacy was 0.94 and Bartlett's Test of Sphericity was significant ($P < 0.001$), in line with published

recommendations[97]. More detailed results of the EFA can be found in Supplementary Information under 'In-lab MEG sample factor analysis'.

### Behavioural analysis

**Smartphone population sample.** We correlated the primary index of information gathering, namely, the number of draws to a decision, as well as accuracy, with the OCI-R scores using Spearman's correlations.

We then investigated the underlying processes of information gathering by fitting participants' draw-wise binary choice to sample information or not to a logit-linked binomial general linear mixed model (GLMM) predicting the probability to make a decision from the experimental factors, OCI-R scores and the interactions of OCI-R scores with the experimental factors, as well as random intercepts and slopes for each experimental factor.

To quantify the evidence strength (ES), we define the following variables. The cumulative ES for stimulus A at a given draw is the cumulative sum of instances of A over all draws, for example, all yellow gems seen until the current draw d:

$$ES_{d,A} = \sum_d N_A \tag{1}$$

and analogously for stimulus B:

$$ES_{d,B} = \sum_d N_B \tag{2}$$

We define the evidence strength for the current majority stimulus, that is, that which is more abundant on the current draw, $ES_{d,majority}$ as $ES_{d,A}$ when $ES_{d,A} > ES_{d,B}$ and as $ES_{d,B}$ when $ES_{d,B} > ES_{d,A}$. Vice versa, $ES_{d,minority}$ is defined as $ES_{d,A}$ when $ES_{d,A} < ES_{d,B}$ and as $ES_{d,B}$ when $ES_{d,B} < ES_{d,A}$.

The total evidence strength at a given draw is defined as the difference between the evidence for the stimulus in the current majority and that for the stimulus in the current minority:

$$ES_{d,total} = ES_{d,majority} - ES_{d,minority} \tag{3}$$

Note that $ES_{d,total}$ is therefore by definition always positive.

$ES_{d-1}$ is thus the cumulative relative evidence strength for the current majority stimulus until the previous draw d−1. Finally, we define the update in evidence strength as

$$\Delta ES_d = ES_{d,total} - ES_{d-1,total} \tag{4}$$

Thus, experimental factors consisted of the cumulative evidence strength on the previous draw $ES_{d-1}$, the update in evidence strength $\Delta ES$, and the current number of draws $N_d$.

The GLMM was thus defined as

$$p(\text{decide}) \approx OCIR \times N_d + OCIR \times \Delta ES + OCIR$$
$$\times ES_{d-1} + (1 + N_d + \Delta ES + ES_{d-1}|\text{participant}) \tag{5}$$

with a logit link function (main effects were included in the analysis but are omitted here for legibility).

In a follow-up analysis, we tested the recency effect suggested by these results in a more targeted manner by fitting a GLMM predicting $p(\text{decide})$ from the current evidence at draw d, d−1 and d−2:

$$p(\text{decide}) \approx ES_{d,total} + ES_{d-1,total} + ES_{d-2,total}$$
$$+ (1 + ES_{d,total} + ES_{d-1,total} + ES_{d-2,total}|\text{participant}) \tag{6}$$

with a logit link function, whereby the evidence $ES_{d,total}$ is defined relative to the chosen option as opposed to the current majority:

$$ES_{d,total} = ES_{d,chosen} - ES_{d,unchosen} \tag{7}$$

To corroborate the individual differences we found in the recency effect associated with OC symptoms, we derived a metric of recency

bias for each participant by fitting a separate generalized linear model (GLM) to each participant analogous to the GLMM

$$p(\text{decide}) \approx ES_{d,total} + ES_{d-1,total} + ES_{d-2,total} \tag{8}$$

with a logit link function, and taking the relative difference between the beta coefficients for $ES_{d,total}$ and $ES_{d-1,total}$. We then correlated this index of recency bias with the OCI-R score.

Please note that for simplicity, we omit the subscript 'total' from $ES_{d,total}$ and $ES_{d-1,total}$ in the main text and figures.

Lastly, we conducted a mediation analysis evaluating how the observed correlation between OCI-R and sensitivity to current evidence relates to the correlation between OCI-R and confidence ratings, as opposed to the probability to commit to a decision, using the M3 mediation toolbox in MATLAB (https://github.com/canlab/Mediation-Toolbox)[54,55], as in previous work in our research group[98–100]. To perform this analysis, we obtained a measure of sensitivity to $\Delta ES$ in deciding to commit to a choice for each individual participant by fitting a separate GLM to each participant analogous to the GLMM above

$$p(\text{decide}) \approx N_d + \Delta ES + ES_{d-1} \tag{9}$$

with a logit link function.

We then tested whether the individual beta coefficients for $\Delta ES$ mediated the negative association between mean confidence and OCI-R scores. In other words, we take OCI-R as the independent variable, confidence as the outcome variable, and individual beta coefficients for $\Delta ES$ as the mediating variable. The $c$ path represents the total effect of OCI-R on confidence (without controlling for the mediating variable); $c'$ is the direct effect of OCI-R on confidence (controlling for the individual beta coefficients for $\Delta ES$); $b$ is the effect of the individual beta coefficients for $\Delta ES$ on confidence, controlling for OCI-R; $a$ is the effect of OCI-R on the individual beta coefficients for $\Delta ES$; and the product $ab$ is the indirect effect of OCI-R on confidence through the individual beta coefficients for $\Delta ES$. We tested for statistical significance using a bootstrap test with 10,000 bootstrap samples (https://github.com/canlab/MediationToolbox)[54,55].

The rationale for the particular model we tested (OCI-R → $\Delta ES$ → confidence) was motivated by evidence from the field of metacognition showing a directional link between evidence strength and confidence (that is, stronger evidence strength preceding higher confidence ratings, as reviewed recently[101]). Likewise, there is also evidence for low confidence in OCD[53]. With respect to the directionality assumptions, note that here we measure confidence in the restricted context of this particular task as opposed to a trait-like measure of confidence. We therefore asked whether the observed lowered confidence associated with high OCI-R scores is mediated through $\Delta ES$, rather than whether confidence in a task would be causal with respect to a stable mental health trait (OCI-R).

**In-lab MEG sample.** We performed an analogous analysis, while accounting for sample and task differences. As stated above, we used the OC factor score to quantify variability in OC symptoms. In addition, we adapted the GLMM by including the horizon condition and the interactions between the horizon and all other regressors. The interaction of $\Delta ES$ with the horizon was omitted in the participant-level GLMs, as the beta coefficient was not significantly different from zero.

The GLMM was thus defined as:

$$p(\text{decide}) \approx OC \times (N_d + \text{horizon} + \Delta ES + ES_{d-1}) + \text{horizon}$$
$$\times (OC + N_d + \Delta ES + ES_{d-1}) + \text{horizon}$$
$$\times (OC \times N_d + OC \times \Delta ES + OC \times ES_{d-1}) \tag{10}$$
$$+ (1 + N_d + \text{horizon} + \Delta ES + ES_{d-1}$$
$$+ \text{horizon} \times (N_d + \Delta ES + ES_{d-1})\text{participant})$$

with a logit link function (main effects were included in the analysis but are omitted for legibility).

## MEG analysis

**Data acquisition and preprocessing.** Participants sat upright in the scanner while MEG was recorded continuously using a 275-channel system, whereby sensor MLO42 was missing for all participants, and in addition, sensor MRC12 was missing for 59 participants. Empty channels were added in these cases to keep the dimensionality of the MEG data consistent across participants. Bad-quality channels were removed and not interpolated. Physiological and technical artefact removal was performed manually using Fieldtrip. The data was downsampled to 100 Hz and high-pass filtered at 0.5 Hz to remove slow drift effects. Data were segmented into 1,350-ms segments from −100 to 1,250 ms relative to stimulus onset. MEG data were $z$-scored by block and channel (univariate normalization).

**Decoding analysis.** Whole-brain MEG activity was used to decode ΔES using an iterative multivariate lasso regression approach[57,102,103]. Here we estimated the lasso regression 2,000 times, using a random subset of 50 sensors on each iteration. While the method was originally developed to determine which sensors contribute most to the decoding, its primary benefit in this study is that it increases reliability of the individual differences analysis relative to the results from a single iteration of a multivariate lasso regression, by leveraging the many iterations of the lasso (see also Supplementary Information 'Alternative analyses of the localization of the ΔES representation'). The regularization parameter $\lambda$ was set at 0.01 using 5-fold cross-validation. The value for the regularization parameter was tuned by optimizing the cross-validation performance of an orthogonal variable, namely, the current evidence for yellow (correlation between current evidence for yellow and ΔES: $r = 2.755 \times 10^{-4}$, $P = 0.920$), and was fixed for all subsequent whole-brain analyses. The initial optimization procedure was conducted using a non-iterative multivariate lasso regression using all 273 channels, yet note that the decodability using the iterative method remains extremely similar for a range of regularization parameter values in follow up analyses (Supplementary Fig. 4d).

The correlation between the tested and trained data constituted the measure of decodability. Non-parametric cluster-based permutation testing (CBPT[104]) was used to assess significant differences in decodability timecourses. CBPT is commonly used in M/EEG analysis to test for significant differences in a given timeseries while accounting for multiple comparisons (for example, in previous work[105,106]). Briefly, CBPT takes the temporal structure of the data into account and uses permutation tests (here, $N = 1,000$) to build a null distribution of clusters above a given size (here, threshold $t = 3.1$) within the same temporal structure, enabling the evaluation of whether an effect of interest exceeds the expected cluster size under the null.

Individual differences in decodability were assessed by predicting the decodability at each timepoint from the 3 factor scores in individual GLMs. Significant differences in the beta regressor timecourses were assessed by non-parametric CBPT (here, threshold $t = 2.1$). Correlation values were normalized by dividing these by the maximum correlation value for visualization purposes, yet note that all statistical tests were performed prior to normalization.

To investigate which brain areas underpin the decoding, we used a searchlight analysis, such that the lasso regression was estimated 273 times, that is, for each sensor and its direct neighbours, determined using the Fieldtrip 'ft_prepare_neighbours' function. We chose this method instead of the iterative multivariate lasso regression described above for the localization of the effect because we and others[102] have found sensor contribution maps derived from the iterative multivariate lasso regression to be very diffuse (see Supplementary Information 'Alternative analyses of the localization of the ΔES representation'). Given the stark difference in the number of sensors used in the searchlight analysis, we first fine-tuned the value of the regularization parameter $\lambda$ in the same way as above and set it to 0.001. We then calculated the mean decodability per sensor within a given time window of interest to generate sensor contribution maps. To explore individual differences in the sensor contribution maps, we fit a GLM predicting the sensor contribution per sensor averaged across the time window of interest from the 3 factor scores. To keep the sensor subsets constant across participants, sensors MLO42 and MRC12 were omitted for all.

## Reporting summary

Further information on research design is available in the Nature Portfolio Reporting Summary linked to this article.

## Data availability

Behavioural and summary neuroimaging data are publicly available on OSF at https://osf.io/fks97 (ref. 107). Source data are provided with this paper.

## Code availability

Analysis scripts are publicly available on OSF at https://osf.io/fks97 (ref. 107).

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

## Acknowledgements

We thank the members of the Developmental Computational Psychiatry group who participated in a hackathon forming the basis of some of the findings presented here; the Central London OCD support group, the OCD Action charity, the North East London NHS foundation, especially the IAPT services at Waltham Forest, Redbridge and Barking & Dagenham, and the Camden and Islington NHS talking therapies service iCope for help with patient recruitment. T.U.H. was supported by a Sir Henry Dale Fellowship (211155/Z/18/Z; 211155/Z/18/B; 224051/Z/21) from Wellcome and Royal Society, a grant from the Jacobs Foundation (2017-1261-04), the Medical Research Foundation, a 2018 NARSAD Young Investigator grant (27023) from the Brain and Behavior Research Foundation, and a Philip Leverhulme Prize from the Leverhulme Trust (PLP-2021-040). T.U.H. was also supported by the Carl-Zeiss-Stiftung. A Wellcome Trust Cambridge–UCL Mental Health and Neurosciences Network grant (095844/Z/11/Z) supported this work. The Max Planck UCL Centre is a joint initiative supported by UCL and the Max Planck Society. The Wellcome Centre for Human Neuroimaging is supported by core funding from the Wellcome Trust (203147/Z/16/Z). This research was funded in whole, or in part, by the Wellcome Trust (211155/Z/18/Z). For the purpose of open access, the author has applied a CC BY public copyright license to any author-accepted manuscript version arising from this submission.

## Author contributions

M.d.R., N.T. and T.U.H. analysed the data. G.P. and T.U.H. performed the experiments. L.T.H., M.M., R.J.D. and T.U.H. conceived and designed the experiments. All authors wrote the paper.

## Competing interests

T.U.H. consults for Limbic Ltd. and holds shares in the company, which is unrelated to the current project. The other authors declare no competing interests.

## Additional information

**Correspondence and requests for materials** should be addressed to Magdalena del Río or Tobias U. Hauser.

# Reporting Summary

## Statistics

For all statistical analyses, confirm that the following items are present in the figure legend, table legend, main text, or Methods section.

| n/a | Confirmed | |
|---|---|---|
| ☐ | ☒ | The exact sample size (*n*) for each experimental group/condition, given as a discrete number and unit of measurement |
| ☒ | ☐ | A statement on whether measurements were taken from distinct samples or whether the same sample was measured repeatedly |
| ☐ | ☒ | The statistical test(s) used AND whether they are one- or two-sided <br> *Only common tests should be described solely by name; describe more complex techniques in the Methods section.* |
| ☐ | ☒ | A description of all covariates tested |
| ☐ | ☒ | A description of any assumptions or corrections, such as tests of normality and adjustment for multiple comparisons |
| ☐ | ☒ | A full description of the statistical parameters including central tendency (e.g. means) or other basic estimates (e.g. regression coefficient) AND variation (e.g. standard deviation) or associated estimates of uncertainty (e.g. confidence intervals) |
| ☐ | ☒ | For null hypothesis testing, the test statistic (e.g. *F*, *t*, *r*) with confidence intervals, effect sizes, degrees of freedom and *P* value noted <br> *Give P values as exact values whenever suitable.* |
| ☒ | ☐ | For Bayesian analysis, information on the choice of priors and Markov chain Monte Carlo settings |
| ☒ | ☐ | For hierarchical and complex designs, identification of the appropriate level for tests and full reporting of outcomes |
| ☐ | ☒ | Estimates of effect sizes (e.g. Cohen's *d*, Pearson's *r*), indicating how they were calculated |

*Our web collection on statistics for biologists contains articles on many of the points above.*

## Software and code

Policy information about availability of computer code

| Data collection | Smartphone study data was collected using the Brain Explorer app and in-lab study data was collected using MATLAB. |
|---|---|
| Data analysis | Analysis scripts are publicly available at https://osf.io/fks97/ |

For manuscripts utilizing custom algorithms or software that are central to the research but not yet described in published literature, software must be made available to editors and reviewers. We strongly encourage code deposition in a community repository (e.g. GitHub). See the Nature Portfolio guidelines for submitting code & software for further information.

## Data

Policy information about availability of data

All manuscripts must include a data availability statement. This statement should provide the following information, where applicable:

- Accession codes, unique identifiers, or web links for publicly available datasets
- A description of any restrictions on data availability
- For clinical datasets or third party data, please ensure that the statement adheres to our policy

Behavioural and summary neuroimaging data are publicly available at https://osf.io/fks97/.

# Research involving human participants, their data, or biological material

Policy information about studies with [human participants or human data](link). See also policy information about [sex, gender (identity/presentation), and sexual orientation](link) and [race, ethnicity and racism](link).

| | |
|---|---|
| Reporting on sex and gender | In the in-lab sample, sex was self-reported by participants. In the smartphone sample, gender was voluntarily self-reported by participants (i.e., data is not complete). No sex- or gender-based analyses were performed, as no hypotheses were sex- or gender-based. |
| Reporting on race, ethnicity, or other socially relevant groupings | No race, ethnicity, or other socially relevant groupings were used in either the smartphone population or the in-lab sample analysis. |
| Population characteristics | No additional population characteristics are included in the analysis for the smartphone population sample. Potentially relevant population characteristics for the in-lab sample include age, IQ, comorbidities, and medication status - these are not included in any analyses, but are compared across groups and reported in Supplementary Table 2. |
| Recruitment | The smartphone population sample was recruited through public engagement events and word of mouth to use the free app Brain Explorer (www.brainexplorer.net). There is self-selection bias in this sample, as participants are more likely to be engaged in science and/or OCD and mental health fields. The in-lab sample consisted of several groups with different recruitment strategies. OCD and GAD patients were recruited through NHS services, charities and advertisements. To ensure comparable backgrounds, controls were recruited in areas of similar socioeconomic status to the patients'. Participants from the high and low OC groups were recruited from a population-based sample of young people (U-CHANGE study; www.nspn.org.uk) based on their OCD questionnaire score. There is self-selection bias in this sample insofar as participation requires patients to be able and willing to be included, limiting the severity and/or type of symptom profiles represented in the sample. We do not consider any of these biases likely to significantly impact the results. |
| Ethics oversight | University College London (UCL) and NHS research ethics committees |

Note that full information on the approval of the study protocol must also be provided in the manuscript.

# Field-specific reporting

Please select the one below that is the best fit for your research. If you are not sure, read the appropriate sections before making your selection.

☐ Life sciences  ☒ Behavioural & social sciences  ☐ Ecological, evolutionary & environmental sciences

For a reference copy of the document with all sections, see [nature.com/documents/nr-reporting-summary-flat.pdf](nature.com/documents/nr-reporting-summary-flat.pdf)

# Behavioural & social sciences study design

All studies must disclose on these points even when the disclosure is negative.

| | |
|---|---|
| Study description | Data are quantitative - comprising self-report, behavioural and neurophysiological (MEG) data. |
| Research sample | The smartphone population sample is a sample self-selected from the general population, in order to include variability along demographic, cognitive and mental health measures at scale.<br>The in-lab sample consisted of clinically diagnosed OCD and GAD patients, healthy controls and undiagnosed individuals with low and high levels of OCD-like symptoms (as assessed by data available from the U-CHANGE study, see above). The rationale was to include variability along obsessive-compulsive as well as anxiety-related dimensions, relevant to information gathering. |
| Sampling strategy | The smartphone population sample is a convenience sample, self-selected from the general population. The sample size was determined by the population's participation - post hoc sensitivity calculations using G*Power show that the sample of 5,237 is able to detect correlation effects of size 0.04 with 80% power and 0.05 error probability, which is the smallest reported significant effect size. The in-lab sample was selectively sampled. Sample sizes in this case were determined by prior studies as well as pragmatic matters (recruitment and financial limitations). |
| Data collection | The smartphone study was conducted remotely by participants on smartphone devices, such that researchers were not present and no additional details are available.<br>The in-lab study was conducted with the researcher present. During the interviews, participants and researchers used pen and paper and computers to record responses and comments. During the task, participants had electrophysiological neural signals recorded using a MEG scanner and recorded their responses using a MEG-compatible device. Researchers were not blind to the study hypotheses and experimental conditions were constant across participants during data collection. |
| Timing | The smartphone sample data was collected between November 2020 and August 2023. The in-lab sample data was collected between April 2015 and August 2018. |
| Data exclusions | In the smartphone population analysis, 1,927 participants were excluded because of missing questionnaire data. An additional 1,506 participants were excluded based on performance metrics determined by the analysis requirements. Exclusion criteria were not pre-established. |

In the in-lab sample analysis, two participants were excluded because they did not meet the clinical criteria for GAD, one participant was excluded because of comorbid OCD and GAD, one was excluded due to fully missing questionnaire data (pre-established exclusion criteria). One participant was excluded due to performance-related metrics, and five participants were excluded due to technical difficulties and/or poor MEG data quality (not pre-established exclusion criteria).

Non-participation
There is no record of participants dropping out or declining participation for the smartphone study. No participants dropped out of the in-lab study. Participants who declined participation in the in-lab study were not recorded.

Randomization
Participants were not allocated to experimental groups.

# Reporting for specific materials, systems and methods

We require information from authors about some types of materials, experimental systems and methods used in many studies. Here, indicate whether each material, system or method listed is relevant to your study. If you are not sure if a list item applies to your research, read the appropriate section before selecting a response.

## Materials & experimental systems

| n/a | Involved in the study |
|-----|----------------------|
| ☒ ☐ | Antibodies |
| ☒ ☐ | Eukaryotic cell lines |
| ☒ ☐ | Palaeontology and archaeology |
| ☒ ☐ | Animals and other organisms |
| ☒ ☐ | Clinical data |
| ☒ ☐ | Dual use research of concern |
| ☒ ☐ | Plants |

## Methods

| n/a | Involved in the study |
|-----|----------------------|
| ☒ ☐ | ChIP-seq |
| ☒ ☐ | Flow cytometry |
| ☒ ☐ | MRI-based neuroimaging |

## Plants

Seed stocks
*Report on the source of all seed stocks or other plant material used. If applicable, state the seed stock centre and catalogue number. If plant specimens were collected from the field, describe the collection location, date and sampling procedures.*

Novel plant genotypes
*Describe the methods by which all novel plant genotypes were produced. This includes those generated by transgenic approaches, gene editing, chemical/radiation-based mutagenesis and hybridization. For transgenic lines, describe the transformation method, the number of independent lines analyzed and the generation upon which experiments were performed. For gene-edited lines, describe the editor used, the endogenous sequence targeted for editing, the targeting guide RNA sequence (if applicable) and how the editor was applied.*

Authentication
*Describe any authentication procedures for each seed stock used or novel genotype generated. Describe any experiments used to assess the effect of a mutation and, where applicable, how potential secondary effects (e.g. second site T-DNA insertions, mosiacism, off-target gene editing) were examined.*

