## [Peer Review File · Nature Human Behaviour]

Indecision and recency-weighted evidence integration in non-clinical and clinical settings

Corresponding Author: Dr Magdalena del Rio

Version 0:

Decision Letter:

12th February 2025

Dear Dr. del Rio,

Thank you once again for your manuscript, entitled "Indecision and recency-weighted evidence integration in non-clinical and clinical settings", and for your patience during the peer review process.

Your Article has now been evaluated by 3 referees. You will see from their comments copied below that, although they find your work of potential interest, they have raised quite substantial concerns. In light of these comments, we cannot accept the manuscript for publication, but would be interested in considering a revised version if you are willing and able to fully address reviewer and editorial concerns.

We hope you will find the referees' comments useful as you decide how to proceed. If you wish to submit a substantially revised manuscript, please bear in mind that we will be reluctant to approach the referees again in the absence of major revisions. We are committed to providing a fair and constructive peer-review process. Do not hesitate to contact us if there are specific requests from the reviewers that you believe are technically impossible or unlikely to yield a meaningful outcome.

To guide the scope of the revisions, the editors discuss the referee reports in detail within the team, including with the chief editor, with a view to (1) identifying key priorities that should be addressed in revision and (2) overruling referee requests that are deemed beyond the scope of the current study. We hope that you will find the prioritised set of referee points to be useful when revising your study. Please do not hesitate to get in touch if you would like to discuss these issues further.

1. Reviewer 2 raises important concerns regarding the clarity of the results presentation, which may impact their interpretation and understanding. Please carefully revise your manuscript in line with the reviewer's suggestions and consider consulting a statistician for additional guidance.

2. Reviewer 3 requests a more explicit discussion of the ecological validity of your findings and their applicability to clinically diagnosed OCD patients. Please ensure these concerns are thoroughly addressed in your revised manuscript.

3. In your revision, please ensure that key concepts are clearly defined at the beginning of the main text to enhance clarity and reader comprehension.

If you wish to submit a suitably revised manuscript, we would hope to receive it within 4 months. I would be grateful if you could contact us as soon as possible if you foresee difficulties with meeting this target resubmission date.

- Include a "Response to the editors and reviewers" document detailing, point-by-point, how you addressed each editor and referee comment. If no action was taken to address a point, you must provide a compelling argument. When formatting this document, please respond to each reviewer comment individually, including the full text of the reviewer comment verbatim followed by your response to the individual point. This response will be used by the editors to evaluate your revision and sent back to the reviewers along with the revised manuscript.
- Highlight all changes made to your manuscript or provide us with a version that tracks changes.

Link Redacted

Thank you for the opportunity to review your work. Please do not hesitate to contact me if you have any questions or would like to discuss the required revisions further.

Sincerely,

[Redacted]

[Redacted]

Nature Human Behaviour

Reviewer expertise:

Reviewer #1: biases ; OCD

Reviewer #2: magnetoencephalography ; computational psychiatry

Reviewer #3: magnetoencephalography ; computational psychiatry

REVIEWER COMMENTS:

Reviewer #1 (Remarks to the Author):

Thank you for offering me to review this manuscript.

I structured my review according to the journals guidelines for review:

1. Key results: Please summarize what you consider to be the outstanding features of the work.

- I see several strengths in the paper: 1. Two large samples ($n > 5,000$ crowd-source and $n > 100$ in lab-based [clinical]sample); 2. Mixture of methods (behavioral and neuroimaging), 3. Transdiagnostic perspective in both theory and methodologically, 4. Good visualizations.

2. Validity: Does the manuscript have flaws which should prohibit its publication? If so, please provide details.

- The authors have conducted a comprehensive study including two samples, usage of various advanced methodologies (neuroimaging, complex investigating/modelling of behavioral data), embedment in theoretical concepts (especially transdiagnostic). I could not find any fatal flaws.

3. Originality and significance: What are the major claims of the paper? Do you think that they represent a significant advance in the field? If the conclusions are not original, please provide relevant references. On a more subjective note, do you feel that the results presented are of immediate interest to many people in your own discipline, and/or to people from several disciplines?

- The authors investigated indecisiveness, a field which receives increasing attention. The authors laid very much the ground in this project with the study design (the information gathering task). It is both novel and yet robust (e.g. replicated in two different samples). In parts, the paper is hard to read and to follow, so I recommend to revise it according to the points I outline in section 10 (suggested improvements). I think the author managed well to provide most information of methods and results concise in the figures (even though there are also a lot of them and to put some of them in the appendix should be considered).

4. Note that Nature Human Behaviour publishes manuscripts that represent a significant advance in one or more of the following categories:

Conceptual novelty

Methodological novelty

Applied/Societal-/Policy-related Advance

Evidence-based advance [Although a manuscript may lack conceptual novelty, it may represent an evidence-based advance if its scale and/or rigour supersede the existing literature and significantly strengthen confidence in a scientific finding or convincingly falsify it.]

- The paper provides significant advances in term of evidence-based advancement (e.g. two large samples, inclusion of MEG data). Also. the methods and concepts are rather new (task has been used in variations before, however, the link to indecisiveness in the context of OCD from a transdiagnostic perspective has been strengthened).

5. Data & methodology: Please comment on the validity of the approach, quality of the data and quality of presentation. Please note that we expect our reviewers to review all data, including any extended data and supplementary information. Is the reporting of data and methodology sufficiently detailed and transparent to enable reproducing the results?

- The methods used seem very appropriate. The sample sizes of the two studies are large ($n > 5,000$ for online sample; $n > 100$ for laboratory clinical sample), thus the study seems well powered. However, I did not find a formal power calculation.

- The study procedure is well described, including graphical illustration of the task. Thus, a replication should be possible. The only

minor concern might be that the authors might want to have a more formal introduction of the various scores (e.g. confidence, draws to decision) used. Even though they become clear, a concise definition could help to avoid misunderstandings.

- The authors provide very informative, well-designed plots (e.g. displayed distributions for group comparisons, all data points for regression analyses), which is a strength of the manuscript.

6. Preregistration: If any part of the work reported in the manuscript was pre-registered, did the authors follow their preregistration plan? Did they report any deviations from their preregistration? Note that we ask authors to provide a link to the pre-registration in the Methods section and state the date of pre-registration. We also ask that authors disclose all deviations from the pre-registered protocol and explain the rationale for deviation (e.g., flaw, suboptimality, or reviewer/editorial request). In cases of deviation from the analysis plan, the originally planned analyses need to be reported in Supplementary Information.

- The authors do not specify whether the study was pre-registered or not.

7. Appropriate use of statistics and treatment of uncertainties: Please include in your report a specific comment on the appropriateness of any statistical tests, and the accuracy of the description of any error bars and probability values.

- The use of statistical measures seems very appropriate. All figures are described in great depth. The authors did not only provide error bars but also displayed box plots and distribution of data for group comparisons.

8. Custom code: If the work includes custom code, does the code run as intended? If you are unable to access the code, please contact us.

- The authors state: "Data and analysis scripts will be made publicly available on OSF upon publication". I welcome this very much.

9. Conclusions: Do you find that the conclusions and data interpretation are robust, valid and reliable?

- As mentioned before, I find the analyses very robust (two well powered samples; mix of methods). The authors interpret the results in an appropriate manner, both close to the results and both pointing towards various implications.

10. Suggested improvements: Please list additional analyses, experiments or data that could help strengthening the work in a revision.

- In the abstracts you state "high-OC 35 participants show an attenuated neural ES signal", could you explain in the abstract which sample you mean by "high-OC" so that it becomes clearer without reading the entire manuscript.

- In the introduction, I am a bit confused about some wordings in the last paragraph, as you seem to report results already (e.g. "across both studies, we find that participants along an OC spectrum rely less on...", "Using magnetencephalography we show this attenuation") and conclusions already (e.g. "we conclude that indecisiveness along an OC spectrum is driven by a reduced reliance on most recent information."). A slight change in formulations could frame this paragraph clearer as hypotheses.

- I suggest to define scores early on in the methods section, which would make the following sections easier to understand (e.g. you state "mean confidence rating" before explaining confidence in what, the scale etc.)

- I believe there is a small mistake in the methods section: "self-reported symptom severity in patients was assessed using the Y-BOCS interview", should be probably rephrased as "clinician-rated symptom severity".

- Could you please specify how you determined the diagnosis used as exclusion criteria (e.g. autism spectrum disorder or tic disorder).

- I like the figures. The only comment I have is to explain ES a little bit more in figure 1 and 2 (abbreviation used in the subplot headings, but not so clearly defined in the notes).

- Even though I like the transdiagnostic approach of the paper, the authors might be a little bit clearer at points. For example, you state that "indecisiveness in depression and dependent personality disorder has previously been highlighted", however, the reference you are citing mainly refers to depression. Potentially, you want to expand the sections on the transdiagnostic perspective including theoretical assumptions or stay closer to empirical literature. I think this could be achieved by minor adjustments.

- In my opinion, the conclusion paragraph could be more on point, e.g. say clearer what your specific finding implicate (rather than saying that it may bring us closer to understanding the mechanisms, explain how it helped us understanding the mechanisms).

- Please make sure everything is formatted correctly (e.g. statistical letters in Italics)

- Some wordings might be adjusted to the journal style (e.g. due to the citation style, "though see also 16,17" might be changed to something like "though others have criticized", line 420).

11. References: Does this manuscript reference previous literature appropriately? If not, what references should be included or excluded?

- Referencing seems very appropriate. My only suggestion is mentioned above (specify the link between ref 76 and indecisiveness in depended PD).

12. Clarity and context: Is the abstract clear, accessible? Are abstract, introduction and conclusions appropriate?

- Despite my two minor comments on the abstract (definition of high-OC) and conclusion paragraph (one more sentence on the specific findings) I find the abstract, introduction and conclusion appropriate and have nothing to add.

Reviewer #2 (Remarks to the Author):

This study reports that the most recent changes in evidence drive decisions in a time-limited decision task. This recency effect is diminished in obsessive compulsive disorder (OCD), and more broadly, correlates with OC ratings even in a normal population and in a population with anxiety disorder. The study does a nice job in replicating results in a large online cohort. In the lab, they argue that MEG signals reflect their metric of change in evidence, which they refer to as "evidence strength updates".

Unfortunately this term and the definition of evidence accumulation are very poorly explained in this paper. This is unfortunate, because it is a central concept they would like to promote in this work by associating it with various behavioral measures, OCD phenotype and neural activity.

I am not an expert in decision making or OCD so can not speak to the novelty of the result or if the introduction and Discussion are complete and balanced. On the surface they appear well researched and are clearly written. I do see major problems however with the presentation of their statistical analysis, behind which all manner of problems could be lurking. Nothing, however, that a careful response could not address. I do recommend to have a *different* a person looks at the sections in question, perhaps someone with a more math/stats background in the existing team.

Here some detailed comments:

$|\sum ES_{d-1}|$ and ΔES are never properly defined. All we are given is a single figure panel 1A that I do not manage to understand. In that example it seems that "ES" are +1 or -1 depending on which gem appears on the uncovered location. And $|\sum ES_{d-1}|$ stands for the absolute value of the sum from $i=0$ to $i=d-1$. So $|\sum_{i=0}^{d-1} ES_d|$. In that example of Fig 1A I was not able to figure out what ΔES stands for. I see [-,-1,1,1,-1] but I have no idea how that matches either the absolute sum of the gems showing, which are [y,b,b,b,y] (y=yellow, b=blue). Given that this "evidence strength updates" is the most important concept in the paper, it would help to have a clear definition please.

When I look at equations (1) and (2) in the Methods, I am only slightly more clear. Equation (1) seems to suggest that $ES_{\{A,i\}}$ and $ES_{\{B,i\}}$ for the i -th draw are {0,1} variables depending on what the outcome of a draw was. Cumbersome way to write this, but OK. But what the hell is that < sign doing there? Did you mean to say that there is a sum that ends with $d-1$? Then I look at Equation (2) and am really confused. It seems this is the pairwise difference across draws, so basically the quantity that was being integrated, i.e. the value of the gems {-1,+1}. In the example in the figure that would mean [-1,1,1,1,-1] or alternatively [-,-1,1,1,1] depending on the indexing. Neither is shown in Fig 1A. And again, there is this mysterious < sign. The text underneath "the difference in relative evidence in draw d $|\sum ES_d|$ and the previous draw $|\sum ES_{d-1}|$ does not help. The equation has no absolute value $||$ symbols while the text does. So which is it? All this strikes me as pseudo math. Utterly confusing. I suggest you have a math-inclined person suggest less confusing wording and notation.

After looking at Fig 1A for a long long time. I think I get it now, but still think there is something wrong with the figure. The first gem gives evidence for yellow. The sum and the gain of information should both be 1, the next shows blue, the sum should be 0 and the gain -1, the next is blue, we should have sum 1 and gain 1. It seems that this is almost what you have for gain (except the first – should be +1). and the sum is shifted by 1 sample. I understand why you want to shift, but then you should shift the gain as well along with it. Currently one is shifter and the other is not. I wonder how you actually did the regression, with one shifted and the other not? Does not seem correct to me.

"Model comparison favoured the aforementioned distinction over a model where all evidence was aggregated in a single predictor ($\Delta AIC = 1.709 \times 10^6$; $\Delta BIC = 1.709 \times 10^6$)" I have no idea what this means, as neither model was specified. Please be specific. The reader should not need to go to the methods to know what your result means.

Checking the methods I read "fitting participants' draw-wise binary choice to sample information or not to a logit-linked binomial general linear mixed model (GLMM) predicting the probability to make a decision from the experimental factors, OCI-R scores, and the interactions of OCI-R scores with the experimental factors, as well as random intercepts and slopes for each experimental factor." Again, I have no idea what was done. Correct me if I am wrong, I would expect something like this in Wilkinson notation:

decision ~ OCI-R + delta_ES + abs_sum_ES + OCI-R*delta_ES + OCI-R*abs_sum_ES + (1|subject)

with a logistic link function. If this is what you did, then write it in the methods. If you did something else, then be explicit so we know.

Then I read further in the methods “ Lastly, we conducted a mediation analysis evaluating the relationship between confidence, OCI-R and sensitivity to current evidence ... to perform this analysis, we obtained a measure of sensitivity to ΔES for each individual participant by fitting ... based on the same experimental factors.” What? How is confidence part of the analysis, yet you are using the same factors as before, where confidence was not a factor? And what do you mean by sensitivity? Do you mean how the beta coefficient for ΔES varies by subject? Should that not be done by adding a random effect + ($\Delta ES|subject$) to the mixed effect model.

I generally know what mediation analysis is, and it does not require a toolbox, all you need is to apply two models, one for the dependent variable and another to the mediation variable. In this particular case, you seem to suggest: confidence $\rightarrow \Delta ES \rightarrow$ OCR-score with a possible direct effect confidence \rightarrow OCR-score. But how can you assume this direction of arrows? Could it not equally well be $\Delta ES \rightarrow$ confidence \rightarrow OCR-score or any other direction of effects. I don't think there is strong a priori justification for such causal assumptions as are needed for mediation analysis of these 3 variables. Actually, I now see your figure 1F, that has confidence as the dependent (outcome) variable. I would have expected lack of confidence is one of the causes for OCD, but maybe not. All this is arbitrary and only supports the concern that mediation analysis only makes sense when there are strong a priori reasons to rule out particular direction for some of the possible arrows between 3 variables.

Figure 1C seems to indicate that # draws were also used. This should be said in the text. This figure shows on the vertical axes $p(\text{decision})$. How can a probability be negative? Or larger than 1? you probably mean to write “logistic regression coefficients” or “beta coefficients”, or something like that.

“Next, we investigated which of the identified cognitive contributors explained differences in task behaviour in high-OC participants.” It sounds like you are running first one model, then another, and then another. But in figure 1C you seem to show beta coefficients for all variables you considered. The way this should be done is to decide ahead of time on all possible factors and run a single model. Otherwise you get into horrible multiple comparison problems. What did you actually do when reporting all these p-values up to this point? Single model running one variable at a time?

“We found that OC symptoms were linked to reduced weighting of previous evidence $|ES_{d-1}|$ ($= -0.053$, $SE = 0.007$, $p < 0.001$) and particularly of ES ($= -0.085$, $SE = 0.009$, $p < 0.001$).” You probably mean the interaction terms shown in figure 3C? If so, say so.

“Further, by fitting an additional GLM predicting $p(\text{decide})$ from the current evidence at draw d , $d-1$ and $d-2$ relative to the chosen option per participant, we found that a difference in beta weights for the evidence at draw d and the evidence at draw $d-1$ correlated negatively with the OCI-R score ($s = -0.068$, $p < 0.001$), supporting the notion of a decreased recency effect.” At this point one really worries about multiple comparisons. Testing all these many models is really not statistically sound. And by the way, the negative beta value says the same as the beta value for delta evidence. I suggest you simply omit this extra analysis.

Figure 4A; “Individual differences in OC factor predict significantly decreased decodability within 420-470 ms and 530-560 ms post stimulus (right y-axis).” I do not understand what the orange curve is, nor what this caption text says.

“We found that ES representations were attenuated in high-OC participants, whereby two time periods reached significance using cluster-based permutation testing, namely 420-470 ms and 530-560 ms after stimulus onset (Fig. 4A-B; controlling for other psychiatric factors).” I do not know what this means or how it was done.

“Neither the decodability of the variable nor the individual differences appear driven by the effect of a commitment to a final decision, as both decodability and individual differences associated with the OC factor remained consistent after omitting the sample immediately preceding the response” I do not understand the rationale of this. What does the previous sample have to do with “commitment”?

While Figure 4B seems reasonably clear, the time window seems horribly cherry picked! The methods section on this is not trust inspiring. “Non-parametric cluster-based permutation tests ...” Given the many doubts above, the remaining language in this paragraph is suspect as it is all rather cryptic. This will need to be explained with more clarity.

“Here, we estimated the lasso regression 2,000 times, using a random subset of 50 sensors on each iteration. The primary benefit of this method is the increased reliability of the individual differences analysis, relative to a single iteration of a multivariate lasso regression using all channels. “ I don’t understand this. Usually one uses cross-validation to test performance. Leaving sensors out seems like a very unusual thing to do? What do you gain from it? Reduce the number of parameters and thus less overstraining? That is what lasso regularization is supposed to do. I have never seen this done, please explain.

“The correlation between the tested and trained data constituted the measure of decodability.” What? What do you mean by “correlation” here? Typically you want to report AUC or accuracy or some such thing on test data.

Minor:

Page 5: “general linear mixed models” this is typically called generalized linear mixed effect model.

Figure 2B: there are 5 groups here, but the caption does not define the acronyms or even say what these groups are.

All the statistics are missing degrees of freedom or sample number. Please add.

References 51 and 52 are not helpful when pointing to a toolbox. I could not find that M3 toolbox.

Reviewer #3 (Remarks to the Author):

This study investigates how biases in information gathering contribute to indecisiveness, particularly in individuals with OC traits and OCD. The authors propose a novel information integration bias, where recently acquired information is overweighted during decision-making, operationalized as evidence strength updates (Δ ES). Using a large, crowd-sourced dataset ($N = 5,237$), the authors find that attenuated Δ ES-weighting is associated with greater indecisiveness across an OC spectrum. This finding is replicated in a lab-based sample ($N = 105$), which includes individuals with OCD and GAD. To explore the neural basis of these biases, the authors use MEG and identify a late neural signal (~ 920 ms post-stimulus) in mediofrontal areas that tracks Δ ES. Importantly, high-OC participants show reduced Δ ES representation in these regions, while other decision-related processes remain intact.

Strengths:

- Large, well-powered sample provides strong statistical support
- Replication in a controlled clinical population strengthens reliability
- Innovative MEG approach identified neural correlates

Major Comments

1. The study uses a large-scale, crowd-sourced sample, but only a small sample reports an OCD diagnosis. How do the findings in self-reported high-OC participants compare to clinically diagnosed OCD patients? If available, a separate analysis of diagnosed OCD participants vs. high-OC but non-diagnosed individuals would strengthen clinical relevance.
2. The decision-making task is highly controlled and involves artificial stimuli (hidden locations and stimuli counts). How well does this task generalize to real-world decision-making scenarios relevant to OCD, such as compulsive checking? Consider discussing ecological validity and whether the observed findings translate to clinical OCD behaviors.
3. The study suggests that attenuated Δ ES processing originates in mediofrontal areas, but the functional role of these areas in OC-related decision-making is not fully explored. The mediofrontal cortex is implicated in error monitoring and cognitive control—could altered conflict monitoring rather than evidence weighting per se explain these results? A comparison with previous neuroimaging findings in OCD would help clarify the functional implications of this attenuation. (e.g., Perera, M. P. N., Mallawaarachchi, S., Bailey, N. W., Murphy, O. W., & Fitzgerald, P. B. (2023). Obsessive–compulsive disorder (OCD) is associated with increased engagement of frontal brain regions across multiple event-related potentials. *Psychological Medicine*, 53(15), 7287–7299.)
4. The study demonstrates neural alterations in information processing in high-OC individuals, but it does not discuss whether these findings translate to clinically diagnosed OCD patients.
5. Could these results inform biomarker development for OCD or potential neuromodulation targets (e.g., tACS or TMS over mediofrontal areas)? Adding a discussion on how these findings may be applied to clinical interventions would enhance the study’s translational value.

Minor Comments

1. In Pg 18, methods: the authors talk about two studies ('in both studies'), but it is not clear what the two studies exactly are. Please consider re-writing in two sections (e.g., study 1, study 2)
2. It would be helpful if a PRISMA chart was included in the paper to show how participant recruitment and exclusion occurred.
3. Pg 19, "Group demographics, medication percentages, and comorbidities for the final sample included in the analysis are reported in Supplementary Table 1." It appears to be in Table 2 within the supplementary material.
4. The discussion section could be improved by adding a "Limitations and Future Directions" sub-section.

Version 1:

Decision Letter:

Our ref: NATHUMBEHAV-24125015A

22nd August 2025

Dear Dr. del Rio,

Thank you for submitting your revised manuscript "Indecision and recency-weighted evidence integration in non-clinical and clinical settings" (NATHUMBEHAV-24125015A). It has now been seen by the original referees, as well as a newly recruited reviewer (Reviewer 4), and their comments are below. As you can see, the reviewers find that the paper has improved in revision. We will therefore be happy in principle to publish it in Nature Human Behaviour, pending minor revisions to satisfy the referees' final requests and to comply with our editorial and formatting guidelines.

We are now performing detailed checks on your paper and will send you a checklist detailing our editorial and formatting requirements within two weeks. Please do not upload the final materials and make any revisions until you receive this additional information from us.

Sincerely,

[REDACTED]
[REDACTED]
[REDACTED]
Nature Human Behaviour

Reviewer #3 (Remarks to the Author):

The points raised have been addressed sufficiently.

Reviewer #4 (Remarks to the Author):

This is an important study that bridges population-level computational psychiatry with in-lab brain imaging. The authors identify attenuated weighting of recent evidence (Δ ES) as a computational mechanism linked to indecisiveness and obsessive-compulsive symptoms. The combination of very large online samples, replication in a clinical cohort, and convergent MEG data is rare in the field and gives the findings unusual strength.

The evidential base is particularly strong. More than 5,000 online participants provide statistical power and generalizability, and the key findings are reproduced in a clinical sample including OCD and GAD patients. The MEG analyses then reveal a temporally distinct neural signal of Δ ES in medial frontal regions, attenuated in high-OC individuals. Taken together, the convergence across behavioral, clinical, and neural data sets a high standard for explanatory strength.

Concerns raised during the first round of review have been addressed thoroughly. Questions about statistical robustness are resolved by the addition of preregistration details, full model comparison tables, cross-validation, and parameter recovery analyses. These show that the Δ ES parameter is stable and not the result of overfitting or selective reporting. Likewise, concerns about the neural evidence have been met with convergent source reconstructions and more cautious framing. The late Δ ES signal, appearing around 920 ms and present even on correct trials, is distinct from error-monitoring. Although MEG localization inevitably carries uncertainty, the convergence of methods and the circumspect interpretation are sufficient to support the authors' claims.

The authors themselves note that their paradigm is optimized to isolate evidence accumulation and does not capture all forms of indecision seen in daily life. I agree this is an important limitation and would suggest foregrounding it slightly more in the discussion, so that readers appreciate both the strength of the mechanistic isolation and the limits of ecological generalization. Presentation could also be improved by clearer cross-referencing between main figures and supplementary analyses, which would help readers navigate the technical results.

Overall, this is an ambitious and carefully executed piece of work. The revision has strengthened the manuscript considerably, and in my judgment all substantive concerns have now been resolved. I see no reason to delay publication further and recommend acceptance after only minor editorial adjustment.

Version 2:

Decision Letter:

Dear Dr del Rio,

We are pleased to inform you that your Article "Indecision and recency-weighted evidence integration in non-clinical and clinical settings", has now been accepted for publication in Nature Human Behaviour.

Authors may need to take specific actions to achieve compliance with funder and institutional open access mandates. If your research is supported by a funder that requires immediate open access (e.g. according to [Plan S principles](https://www.springernature.com/gp/open-science/plan-s-compliance) or the [NIH public access policy](https://www.springernature.com/gp/open-science/us-federal-agency-compliance)) then you should select the gold OA route, and we will direct you to the compliant route where possible. Because authors warrant under our subscription licensing terms that they haven't committed to licensing any version of their article under a licence inconsistent with the terms of our agreement – including the applicable embargo period – publication under the subscription model isn't suitable for authors whose funders require no embargo.

With best regards,

[Redacted signature]

[Redacted signature]

Nature Human Behaviour

P.S. Click on the following link if you would like to recommend Nature Human Behaviour to your librarian
<http://www.nature.com/subscriptions/recommend.html#forms>

** Visit the Springer Nature Editorial and Publishing website at http://editorial-jobs.springernature.com?utm_source=ejp_NHumB_email&utm_medium=ejp_NHumB_email&utm_campaign=ejp_NHumB for more information about our career opportunities. If you have any questions please click [here](mailto:editorial.publishing.jobs@springernature.com). **

Summary

1. Reviewer 2 raises important concerns regarding the clarity of the results presentation, which may impact their interpretation and understanding. Please carefully revise your manuscript in line with the reviewer's suggestions and consider consulting a statistician for additional guidance.
2. Reviewer 3 requests a more explicit discussion of the ecological validity of your findings and their applicability to clinically diagnosed OCD patients. Please ensure these concerns are thoroughly addressed in your revised manuscript.
3. In your revision, please ensure that key concepts are clearly defined at the beginning of the main text to enhance clarity and reader comprehension.

R0.1: We thank the editor for their guidance and for the summary of the key suggestions. We have now implemented all of the above, as well as addressed all detailed reviewer comments below.

We believe that our revised version addresses all of the concerns raised and we also ensured it adheres to your journal's requirements.

Reviewer expertise:

Reviewer #1: biases ; OCD

Reviewer #2: magnetoencephalography ; computational psychiatry

Reviewer #3: magnetoencephalography ; computational psychiatry

REVIEWER COMMENTS:

Reviewer #1

Thank you for offering me to review this manuscript.

I structured my review according to the journals guidelines for review:

1. Key results: Please summarize what you consider to be the outstanding features of the work.

- I see several strengths in the paper: 1. Two large samples ($n > 5,000$ crowd-source and $n > 100$ in lab-based [clinical]sample); 2. Mixture of methods (behavioral and neuroimaging), 3. Transdiagnostic perspective in both theory and methodologically, 4. Good visualizations.

2. Validity: Does the manuscript have flaws which should prohibit its publication? If so, please provide details.

- The authors have conducted a comprehensive study including two samples, usage of various advanced methodologies (neuroimaging, complex investigating/modelling of behavioral data), embedment in theoretical concepts (especially transdiagnostic). I could not find any fatal flaws.

3: Originality and significance: What are the major claims of the paper? Do you think that they represent a significant advance in the field? If the conclusions are not original, please provide relevant references. On a more subjective note, do you feel that the results presented are of immediate interest to many people in your own discipline, and/or to people from several disciplines?

- The authors investigated indecisiveness, a field which receives increasing attention. The authors laid very much the ground in this project with the study design (the information gathering task). It is both novel and yet robust (e.g. replicated in two different samples). In parts, the paper is hard to read and to follow, so I recommend to revise it according to the points I outline in section 10 (suggested improvements). I think the author managed well to provide most information of methods and results concise in the figures (even though there are also a lot of them and to put some of them in the appendix should be considered).

4. Note that Nature Human Behaviour publishes manuscripts that represent a significant advance in one or more of the following categories:

Conceptual novelty, Methodological novelty, Applied/Societal-/Policy-related Advance

Evidence-based advance [Although a manuscript may lack conceptual novelty, it may represent an evidence-based advance if its scale and/or rigour supersede the existing literature and significantly strengthen confidence in a scientific finding or convincingly falsify it.]

- The paper provides significant advances in term of evidence-based advancement (e.g. two large samples, inclusion of MEG data). Also. the methods and concepts are rather new (task has been used in variations before, however, the link to indecisiveness in the context of OCD from a transdiagnostic perspective has been strengthened).

5. Data & methodology: Please comment on the validity of the approach, quality of the data and quality of presentation. Please note that we expect our reviewers to review all data, including any extended data and supplementary information. Is the reporting of data and methodology sufficiently detailed and transparent to enable reproducing the results?

- The methods used seem very appropriate. The sample sizes of the two studies are large ($n > 5,000$ for online sample; $n > 100$ for laboratory clinical sample), thus the study seems well powered. However, I did not find a formal power calculation.

- The study procedure is well described, including graphical illustration of the task. Thus, a replication should be possible. The only minor concern might be that the authors might want to have a more formal introduction of the various scores (e.g. confidence, draws to decision) used. Even though they become clear, a concise definition could help to avoid misunderstandings.

- The authors provide very informative, well-designed plots (e.g. displayed distributions for group comparisons, all data points for regression analyses), which is a strength of the manuscript.

6. Preregistration: If any part of the work reported in the manuscript was pre-registered, did the authors follow their preregistration plan? Did they report any deviations from their preregistration? Note that we ask authors to provide a link to the pre-registration in the Methods section and state the date of pre-registration. We also ask that authors disclose all deviations from the pre-registered protocol and explain the rationale for deviation (e.g., flaw, suboptimality, or reviewer/editorial request). In cases of deviation from the analysis plan, the originally planned analyses need to be reported in Supplementary Information.

- The authors do not specify whether the study was pre-registered or not.

7. Appropriate use of statistics and treatment of uncertainties: Please include in your report a specific comment on the appropriateness of any statistical tests, and the accuracy of the description of any error bars and probability values.

- The use of statistical measures seems very appropriate. All figures are described in great depth. The authors did not only provide error bars but also displayed box plots and distribution of data for group comparisons.

8. Custom code: If the work includes custom code, does the code run as intended? If you are unable to access the code, please contact us.

- The authors state: "Data and analysis scripts will be made publicly available on OSF upon publication". I welcome this very much.

9. Conclusions: Do you find that the conclusions and data interpretation are robust, valid and reliable?

- As mentioned before, I find the analyses very robust (two well powered samples; mix of methods). The authors interpret the results in an appropriate manner, both close to the results and both pointing towards various implications.

10. Suggested improvements: Please list additional analyses, experiments or data that could help strengthening the work in a revision.

R1.0: We appreciate the reviewer's positive assessment of our manuscript, highlighting its novelty, methodological rigour and potential impact. We have thoroughly revised all aspects of the manuscript, addressing the concerns raised above and in the subsequent comments. We believe the reviewers' comments substantially improve our paper and render it more readable and accessible.

Q1.1: In the abstracts you state "high-OC 35 participants show an attenuated neural ES signal", could you explain in the abstract which sample you mean by "high-OC" so that it becomes clearer without reading the entire manuscript.

R1.1: We apologise for this omission. We now refer to high obsessive-compulsive throughout the abstract and prior to the first introduction of the term "OC". Moreover, we have clarified that this finding transcends diagnostic labels.

P. 2, ll. 6-15: "In a large, crowd-sourced sample (N = 5,237), we find a reduced Δ ES-weighting drives indecisiveness along an obsessive-compulsive spectrum. We replicate this attenuated Δ ES-weighting in a second lab-based study (N = 105) that included a transdiagnostic obsessive-compulsive spectrum encompassing OCD and generalised anxiety patients. Using magnetoencephalography (MEG), we trace Δ ES signals to a late neural signal peaking at around 920 ms. Critically, highly obsessive-compulsive participants, across diagnoses, show an attenuated neural Δ ES signal in mediofrontal areas while other decision-relevant processes remain intact. Our findings establish biased information-weighting as a key driver of information gathering, where attenuated Δ ES is linked to indecisiveness across an obsessive-compulsive spectrum."

Q1.2: In the introduction, I am a bit confused about some wordings in the last paragraph, as you seem to report results already (e.g. "across both studies, we find that participants along an OC spectrum rely less on...", "Using magnetencephalography we show this attenuation") and conclusions already (e.g. "we conclude that indecisiveness along an OC spectrum is driven by a reduced reliance on most recent information."). A slight change in formulations could frame this paragraph clearer as hypotheses.

R1.2: We agree. We have now re-formulated it to distinguish hypotheses from pre-emptive results.

P. 3 ll. 31-32 - p. 4 l. 1): *“In this study, we investigate the neurocognitive contributors to information gathering across two large samples and tasks which we previously collected in the lab. To pre-empt the results, we find that [...]”*

Q1.3: I suggest to define scores early on in the methods section, which would make the following sections easier to understand (e.g. you state “mean confidence rating” before explaining confidence in what, the scale etc.)

R1.3: Thank you for pointing this out. We revised the beginning of the manuscript to ensure all technical terms/measures are appropriately introduced in the methods section.

P. 20, ll. 10-12: *“Classically, the key metric in this task is ‘draws to a decision’, i.e., the number of draws or samples the participant chooses to view before deciding for either stimulus.”*

P. 20, ll. 21-23: *“[...] inclusion criteria, which comprised a mean number of draws between 2 and 23 over a 15-trial game, accuracy in the binary choice between the two stimuli above 80%, and a minimum of three unique values in the total set of 15 numbers of draws to a decision [...]”*

P. 20, ll. 25-29: *“Additional exclusion criteria were applied to conduct a further analysis on participants’ confidence ratings in their binary choice between the two possible stimuli, which they provided immediately after each decision on a slider ranging from 0 to 100. These criteria comprised a mean confidence rating over a 15-trial game between 15 and 98 and a minimum of three unique values in the total set of 15 confidence ratings [...]”*

Q1.4: I believe there is a small mistake in the methods section: “self-reported symptom severity in patients was assessed using the Y-BOCS interview”, should be probably rephrased as “clinician-rated symptom severity”.

R1.4: We apologise for this oversight. The Y-BOCS was indeed a clinical interview. We have corrected this mistake.

P. 21, ll. 12-13: *“[...] the Y-BOCS interview¹⁵ (which assesses clinician-rated symptom severity, only for OCD patients) [...]”*

Q1.5: Could you please specify how you determined the diagnosis used as exclusion criteria (e.g. autism spectrum disorder or tic disorder).

R1.5: We apologise for this omission. A majority of participants in the MEG study were recruited through NHS clinics, where the clinicians pre-screened patients to meet the inclusion and

exclusion criteria. Potential participants were subsequently screened by a member of research staff, focusing on existing diagnoses (and MEG inclusion criteria). Eligible participants were then invited to an extensive clinical interview (SCID, Y-BOCS, clinical questionnaires) and results were discussed among the research group to reach a consensus as to whether a participant would, or would not, be included. We specify this further in the revised methods section:

P. 21, ll. 1-19: *“In-lab MEG sample: 115 participants completed the MEG study (data collected between April 2015 and August 2018). OCD and GAD patients were recruited through NHS services, charities and advertisements. To ensure comparable backgrounds, controls were recruited in areas of similar socioeconomic status to the patients. Exclusion criteria were current use of antipsychotic medication, severe learning disability, and comorbidities including current psychosis or bipolar disorder, substance abuse disorder and personality disorders other than obsessive-compulsive personality disorder, as well as self-disclosed autism spectrum, tic or Tourette disorder. As part of the recruitment process through NHS clinics, clinicians pre-screened potential participants based on the inclusion criteria. Those considered eligible were subsequently screened by a study staff member, focusing on existing diagnoses and MEG inclusion criteria. All participants underwent an extensive clinical interview, conducted by experienced researchers at UCL, comprising a structured clinical interview (SCID)⁸³, the Y-BOCS interview¹⁵ (which assesses clinician-rated symptom severity, only for OCD patients), and seven questionnaires, namely the Barratt Impulsiveness Scale (BIS⁸⁴), Beck Depression Inventory II (BDI-II⁸⁵), Intolerance of Uncertainty Scale (IUS⁸⁶), OCI-R⁵¹, PI-WSUR⁸⁷, State and Trait Anxiety Inventory (STAI⁸⁸), and Frost Multidimensional Perfectionism Scale (FMPS⁸⁹). The final decision to include a participant was based on discussion of the results of the interview among the research group. All participants with OCD and GAD fulfilled the corresponding diagnostic criteria for only one of these disorders¹².”*

Q1.5: I like the figures. The only comment I have is to explain ES a little bit more in figure 1 and 2 (abbreviation used in the subplot headings, but not so clearly defined in the notes).

R1.5: Thank you. We now add a more detailed definition of measures of interest, as well as the relevant abbreviation.

Figure 1: *“Here evidence strength (ES) at draw d is quantified as the cumulative difference in evidence for the two gems with respect to the gem which is more abundant at draw d (e.g., here 3 diamonds minus 2 yellow gems constitutes an evidence strength of 1 at draw 5). Cumulative evidence strength at the previous draw ES_{d-1} is defined as the lagged ES by one draw and evidence strength update ΔES is quantified as the signed difference between the cumulative ES at draw $d-1$ and draw d .”*

Figure 2: *“Here again, evidence strength (ES) at draw d is quantified as the cumulative difference in evidence for the two gems with respect to the gem which is more abundant at draw d (e.g., here*

4 blue gems minus 1 yellow gem constitutes an evidence strength of 3 in the first draw, and 8 yellow gems minus 7 blue gems yields an evidence strength of 1 in the third draw). Cumulative evidence strength in the previous draw ES_{d-1} is defined as the lagged ES by one draw and e evidence strength update ΔES is quantified as the signed difference between the cumulative ES at draw $d-1$ and draw d .”

Q1.6: Even though I like the transdiagnostic approach of the paper, the authors might be a little bit clearer at points. For example, you state that “indecisiveness in depression and dependent personality disorder has previously been highlighted”, however, the reference you are citing mainly refers to depression. Potentially, you want to expand the sections on the transdiagnostic perspective including theoretical assumptions or stay closer to empirical literature. I think this could be achieved by minor adjustments.

R1.6: Thank you for this helpful suggestion. We have now changed the reference to the DSM, as this lists as a possible symptom difficulties with decision making under both disorders, and rephrased the allusion to potential commonalities across additional disorders as follows:

P. 18, ll. 26-29: “Indecisiveness can also be a symptom of both depression and dependent personality disorder¹², yet understanding an across-diagnosis overlap in symptom phenomenology and underlying mechanisms will require future studies that include these populations alongside participants with OCD and GAD.”

Q1.7: In my opinion, the conclusion paragraph could be more on point, e.g. say clearer what your specific finding implicate (rather than saying that it may bring us closer to understanding the mechanisms, explain how it helped us understanding the mechanisms).

R1.7: We thank the reviewer for pointing this out. It also resonates with a suggestion from Reviewer 3 (R3.9). We have thus revised the conclusion and future directions section to state the implications more clearly.

P. 19, ll. 12-17: “In sum, within a large crowd-sourced sample as well as a lab-based sample including clinical patients, we provide evidence of an attenuated weighting of ΔES , both at the behavioural and neural level, linked to transdiagnostic OC symptoms. This novel neurocognitive marker highlights the potential contribution of altered evidence integration to excessive information gathering and indecisiveness across health and mental illness, potentially opening avenues to improved behavioural and brain-based interventions for these pervasive symptoms.”

Q1.8: Please make sure everything is formatted correctly (e.g. statistical letters in Italics)

R1.8: Thanks for pointing out these inconsistencies. They are now corrected throughout the revised manuscript.

Q1.9: Some wordings might be adjusted to the journal style (e.g. due to the citation style, “though see also 16,17” might be changed to something like “tough others have criticized”, line 420).

R1.9: We have now implemented this suggestion and ensured the references are correctly embedded:

P. 17, ll. 24-25: “[...] (*though others have reported fewer draws-to-decision in OCD^{16,17}*).

Q1.10: 11. References: Does this manuscript reference previous literature appropriately? If not, what references should be included or excluded?

- Referencing seems very appropriate. My only suggestion is mentioned above (specify the link between ref 76 and indecisiveness in depended PD).

R1.10: We have now adjusted these references (see above under R1.6).

Q1.11: 12. Clarity and context: Is the abstract clear, accessible? Are abstract, introduction and conclusions appropriate?

- Despite my two minor comments on the abstract (definition of high-OC) and conclusion paragraph (one more sentence on the specific findings) I find the abstract, introduction and conclusion appropriate and have nothing to ad.

R1.11: We thank the reviewer for a positive assessment. We have endeavored to implement all suggestions within the tight word limit imposed by the journal guidelines.

Reviewer #2

Q2.1: This study reports that the most recent changes in evidence drive decisions in a time-limited decision task. This recency effect is diminished in obsessive compulsive disorder (OCD), and more broadly, correlates with OC ratings even in a normal population and in a population with anxiety disorder. The study does a nice job in replicating results in a large online cohort. In the lab, they argue that MEG signals reflect their metric of change in evidence, which they refer to as “evidence strength updates”.

Unfortunately this term and the definition of evidence accumulation are very poorly explained in this paper. This is unfortunate, because it is a central concept they would like to promote in this work by associating it with various behavioral measures, OCD phenotype and neural activity.

I am not an expert in decision making or OCD so can not speak to the novelty of the result or if the introduction and Discussion are complete and balanced. On the surface they appear well researched and are clearly written. I do see major problems however with the presentation of their statistical analysis, behind which all manner of problems could be lurking. Nothing, however, that a careful response could not address. I do recommend to have a *different* person looks at the sections in question, perhaps someone with a more math/stats background in the existing team.

R2.1: We thank the reviewer for their assessment of our manuscript and suggestions to improve the clarity of the paper. We have now thoroughly revised all aspects of the paper to comply. In addition, we have consulted computational and mathematics experts who have provided further feedback on how to improve the statistical and computational aspects of the paper. We believe the revised version is now clearer and more precise.

Here some detailed comments:

Q2.2: $|\sum ES_{d-1}|$ and ΔES are never properly defined. All we are given is a single figure panel 1A that I do not manage to understand. In that example it seems that “ES” are +1 or -1 depending on which gem appears on the uncovered location. And $|\sum ES_{d-1}|$ stands for the absolute value of the sum from $i=0$ to $i=d-1$. So $|\sum_{i=0}^{d-1} ES_d|$. In that example of Fig 1A I was not able to figure out what ΔES stands for. I see $[-,-1,1,1,-1]$ but I have no idea how that matches either the absolute sum of the gems showing, which are $[y,b,b,b,y]$ (y =yellow, b =blue). Given that this “evidence strength updates” is the most important concept in the paper, it would help to have a clear definition please.

R2.2 We apologize for a lack of clarity. We have added the following explanation, including an explicit calculation for depicted examples. We have additionally edited the figures to include the cumulative evidence strength at draw d to facilitate this calculation. Also see R2.3 for the revised equations.

Figure 1: “Here evidence strength (ES) at draw d is quantified as the cumulative difference in evidence for the two gems with respect to the gem which is more abundant at draw d (e.g., here 3 diamonds minus 2 yellow gems constitutes an evidence strength of 1 at draw 5). Cumulative evidence strength at the previous draw ES_{d-1} is defined as the lagged ES by one draw and evidence strength update ΔES is quantified as the signed difference between the cumulative ES at draw $d-1$ and draw d .”

Figure 2: “Here again, evidence strength (ES) at draw d is quantified as the cumulative difference in evidence for the two gems with respect to the gem which is more abundant at draw d (e.g., here 4 blue gems minus 1 yellow gem constitutes an evidence strength of 3 in the first draw, and 8 yellow gems minus 7 blue gems yields an evidence strength of 1 in the third draw). Cumulative evidence strength in the previous draw ES_{d-1} is defined as the lagged ES by one draw and evidence strength update ΔES is quantified as the signed difference between the cumulative ES at draw $d-1$ and draw d .”

Q2.3 When I look at equations (1) and (2) in the Methods, I am only slightly more clear. Equation (1) seems to suggest that $ES_{\{A,i\}}$ and $ES_{\{B,i\}}$ for the i -th draw are $\{0,1\}$ variables depending on what the outcome of a draw was. Cumbersome way to write this, but OK. But **what the hell** is that $<$ sign doing there? Did you mean to say that there is a sum that ends with $d-1$? Then I look at Equation (2) and am really confused. It seems this is the pairwise difference across draws, so basically the quantity that was being integrated, i.e. the value of the gems $\{-1,+1\}$. In the example in the figure that would mean $[-1,1,1,1,-1]$ or alternatively $[-,-1,1,1,1]$ depending on the indexing. Neither is shown in Fig 1A. And again, there is this mysterious $<$ sign. The text underneath “the difference in relative evidence in draw d $|\sum ES_d|$ and the previous draw $|\sum ES_{d-1}|$ does not help. The equation has no absolute value $||$ symbols while the text does. So which is it? All this strikes me as pseudo math. **Utterly confusing**. I suggest you have a math-inclined person suggest less confusing wording and notation.

R2.3 We acknowledge this reviewer’s outspoken notion that our notation was not clear enough or, worse still, “pseudo math”. We have now completely revised this entire section. Critically, based on this reviewer’s comment, we consulted additional computational collaborators with formal training in mathematics, which has led us to reformulate the equations and the notation as follows:

P. 24, ll. 18-30 - p. 25, ll. 1-7: “To quantify the evidence strength (ES), we define the following variables. The cumulative ES for stimulus A at a given draw is the cumulative sum of instances of A over all draws, e.g., all yellow gems seen until the current draw d :

$$ES_{d,A} = \sum_d N_A \quad (1)$$

And analogously for stimulus B:

$$ES_{d,B} = \sum_d N_B \quad (2)$$

We define the evidence strength for the current majority stimulus, i.e., that which is more abundant on the current draw, $ES_{d, \text{majority}}$ as $ES_{d,A}$ when $ES_{d,A} > ES_{d,B}$ and as $ES_{d,B}$ when $ES_{d,B} > ES_{d,A}$. Viceversa, $ES_{d, \text{minority}}$ is defined as $ES_{d,A}$ when $ES_{d,A} < ES_{d,B}$ and as $ES_{d,B}$ when $ES_{d,B} < ES_{d,A}$.

The total evidence strength at a given draw is defined as the difference between the evidence for the stimulus in the current majority and that for the stimulus in the current minority:

$$ES_{d, \text{total}} = ES_{d, \text{majority}} - ES_{d, \text{minority}} \quad (3)$$

Note that $ES_{d, \text{total}}$ is therefore by definition always positive.

ES_{d-1} is thus the cumulative relative evidence strength for the current majority stimulus until the previous draw $d-1$. Finally, we define the update in evidence strength as

$$\Delta ES_d = ES_{d, \text{total}} - ES_{d-1, \text{total}} \quad (4)$$

Thus, experimental factors consisted of the cumulative evidence strength on the previous draw ES_{d-1} , the update in evidence strength ΔES , and the current number of draws N_d .”

Q2.4 After looking at Fig 1A for a long long time. I think I get it now, but still think there is something wrong with the figure. The first gem gives evidence for yellow. The sum and the gain of information should both be 1, the next shows blue, the sum should be 0 and the gain -1, the next is blue, we should have sum 1 and gain 1. It seems that this is almost what you have for gain (except the first – should be +1). and the sum is shifted by 1 sample. I understand why you want to shift, but then you should shift the gain as well along with it. Currently one is shifter and the other is not. I wonder how you actually did the regression, with one shifted and the other not? Does not seem correct to me.

R2.4 This is in fact what we had in mind. We have added the following explanation, which we trust clarifies the rationale.

P. 5, ll. 16-22: “To test whether a recency bias^{41–43} is present in an information gathering context, we distinguished total evidence strength on the previous draw (ES_{d-1} ; absolute cumulative evidence difference across draws [1:d-1] for the current majority) from the most recent change in evidence strength or evidence strength update (ΔES ; Fig. 1A). This is analogous to a distinction between prior expectation and prediction error; i.e., the mismatch between the prior expectation and the

actual observation, which is central to frameworks and models on processes spanning perception through to learning across domains^{44,45} .”

Q2.5 “Model comparison favoured the aforementioned distinction over a model where all evidence was aggregated in a single predictor ($\Delta AIC = 1.709 \times 106$; $\Delta BIC = 1.709 \times 106$)” I have no idea what this means, as neither model was specified. Please be specific. The reader should not need to go to the methods to know what your result means.

R2.5: We apologize and have now revised the wording.

P. 5, ll. 23-24: “*Model comparison favoured the model with ΔES and ES_{d-1} as separate predictors over a model where all evidence was aggregated in ES_d as a single predictor [...]*”

Q2.6 Checking the methods I read “fitting participants’ draw-wise binary choice to sample information or not to a logit-linked binomial general linear mixed model (GLMM) predicting the probability to make a decision from the experimental factors, OCI-R scores, and the interactions of OCI-R scores with the experimental factors, as well as random intercepts and slopes for each experimental factor. “ Again, I have no idea what was done. Correct me if I am wrong, I would expect something like this in Wilkinson notation:

decision ~ OCI-R + delta_ES + abs_sum_ES + OCI-R*delta_ES + OCI-R*abs_sum_ES + (1|subject)

with a logistic link function. If this is what you did, then write it in the methods. If you did something else, then be explicit so we know.

R2.6 Thank you for this constructive suggestion. We have now added the Wilkinson notation for the GLMMs and GLMs.

P. 25, ll. 8-18:

“The GLMM was thus defined as

$$p(\text{decide}) \sim OCI - R \times N_d + OCI - R \times \Delta ES + OCI - R \times ES_{d-1} + (1 + N_d + \Delta ES + ES_{d-1} | \text{participant})$$

with a logit link function (main effects were included in the analysis but are omitted here for legibility).

In a follow-up analysis, we tested the recency effect suggested by these results in a more targeted manner by fitting a GLMM predicting $p(\text{decide})$ from the current evidence at draw d , $d-1$ and $d-2$:

$$p(\text{decide}) \sim ES_{d,\text{total}} + ES_{d-1,\text{total}} + ES_{d-2,\text{total}} \\ + (1 + ES_{d,\text{total}} + ES_{d-1,\text{total}} + ES_{d-2,\text{total}} | \text{participant})$$

with a logit link function, [...]"

P. 25, ll. 22-25: “[...] we derived a metric of recency bias for each participant by fitting a separate generalized linear model (GLM) to each participant analogous to the GLMM

$$p(\text{decide}) \sim ES_{d,\text{total}} + ES_{d-1,\text{total}} + ES_{d-2,\text{total}}$$

with a logit link function, [...]"

P. 26, ll. 9-11: “[...] separate GLM to each participant analogous to the GLMM above:

$$p(\text{decide}) \sim N_d + \Delta ES + ES_{d-1}$$

with a logit link function.”

P. 27, ll. 7-13: “The GLMM was thus defined as

$$p(\text{decide}) \sim OC \times (N_d + \text{horizon} + \Delta ES + ES_{d-1}) + \text{horizon} \\ \times (OC + N_d + \Delta ES + ES_{d-1}) + \text{horizon} \\ \times (OC \times N_d + OC \times \Delta ES + OC \times ES_{d-1}) \\ + (1 + N_d + \text{horizon} + \Delta ES + ES_{d-1} | \text{participant})$$

with a logit link function (main effects were included in the analysis but are omitted for legibility).”

Q2.7 Then I read further in the methods “ Lastly, we conducted a mediation analysis evaluating the relationship between confidence, OCI-R and sensitivity to current evidence ... to perform this analysis, we obtained a measure of sensitivity to ΔES for each individual participant by fitting ... based on the same experimental factors.” What? How is confidence part of the analysis, yet you are using the same factors as before, where confidence was not a factor? And what do you mean by sensitivity? Do you mean how the beta coefficient for ΔES varies by subject? Should that not be done by adding a random effect + (ΔES |subject) to the mixed effect model.

R2.7 Thank you for raising this point. This analysis is investigating a distinct, but related, question on the impact of information gathering processes on confidence, expanding on a substantial literature linking OCD to underconfidence. Thus, in this analysis we ask whether a low mean confidence in the decision associated with high OCI-R scores is mediated by the lower ΔES weighting (also associated with high OCI-R scores). We have clarified this in the revised text.

To expand on the rationale, the confidence in the decision is reported *after* the final decision is made, not prior to each draw. Thus, it is neither possible (as confidence is only available on choice trials, not on all preceding ones) nor logical (given the inverse temporal succession) to include this as a regressor in a GLMM predicting $p(\text{decide})$ on each draw. Instead, to address this

question we avail of an analysis that utilizes summary measures of sensitivity to ΔES (beta coefficient in individual GLMs), confidence (mean confidence ratings) and OCI-R.

P. 26, ll. 3-20: “Lastly, we conducted a mediation analysis evaluating how the observed correlation between OCI-R and sensitivity to current evidence relates to the association between OCI-R and confidence ratings, as opposed to the probability to commit to a decision, using the M3 mediation toolbox in MATLAB (<https://github.com/canlab/MediationToolbox>)^{54,55}. To perform this analysis, we obtained a measure of sensitivity to ΔES in deciding to commit to a choice for each individual participant by fitting a separate general linear model (GLM) to each participant analogous to the GLMM above

$$p(\text{decide}) \sim N_d + \Delta ES + ES_{d-1}$$

with a logit link function.

We then tested if the individual beta coefficients for ΔES mediated the negative association between mean confidence and OCI-R scores. In other words, here OCI-R would be the independent variable, confidence the outcome variable, and the individual beta coefficients for ΔES the mediating variable. The c path represents the total effect of OCI-R on confidence (without controlling for the mediating variable); c' , the direct effect of OCI-R on confidence (controlling for the individual beta coefficients for ΔES); b , the effect of the individual beta coefficients for ΔES on confidence, controlling for OCI-R; a , the effect of OCI-R on the individual beta coefficients for ΔES and the product ab , the indirect effect of OCI-R on confidence through the individual beta coefficients for ΔES .”

Q2.8: I generally know what mediation analysis is, and it does not require a toolbox, all you need is to apply two models, one for the dependent variable and another to the mediation variable. In this particular case, you seem to suggest: confidence \rightarrow ΔES \rightarrow OCR-score with a possible direct effect confidence \rightarrow OCR-score. But how can you assume this direction of arrows? Could it not equally well be ΔES \rightarrow confidence \rightarrow OCR-score or any other direction of effects. I don't think there is strong a priori justification for such causal assumptions as are needed for mediation analysis of these 3 variables. Actually, I now see your figure 1F, that has confidence as the dependent (outcome) variable. I would have expected lack of confidence is one of the causes for OCD, but maybe not. All this is arbitrary and only supports the concern that mediation analysis only makes sense when there are strong a priori reasons to rule out particular direction for some of the possible arrows between 3 variables.

R2.8: We thank the reviewer for asking further clarifications on this. Whilst implementing mediation analyses without a toolbox is indeed possible, our implementation has a number of advantages including allowing conduct of more robust non-parametric statistics (which we report in the revised results together with some additional details about the methods). We have previously

used this toolbox in our research group (e.g., Moses-Payne et al., 2021; Habicht et al., 2022; Chew et al., 2019) and our approach here builds on this prior usage and familiarity.

With respect to the rationale for the particular model we tested (OCI-R → ΔES → confidence, as depicted in Figure 1), this is in fact based on a very strong a priori theoretical rationale. The field of metacognition has drawn on a variety of frameworks (evidence accumulation, Bayesian observers, etc.) to reveal a directional link between evidence strength and confidence (i.e., stronger evidence strength preceding higher confidence ratings, as reviewed e.g., by Fleming, 2024). Likewise, there is prior evidence for low confidence in OCD (e.g., Dar et al., 2022). With regards to the directionality of the effect, our approach asks whether the observed lowered confidence in OCD is mediated through ΔES, rather than asking whether confidence in a task is causing a stable mental health trait (OCI-R). We have now clarified this rationale in the revised text.

P. 7, ll. 23-25: *“Specifically, we found that ΔES partially mediated the influence of OC symptoms on confidence ($ab = 0.01$, $SE = 0.003$, $Z = 3.77$, $p < 0.001$, $c = -0.04$, $SE = 0.018$, $Z = -2.40$, $p = 0.016$, $c' = -0.05$, $SE = 0.018$, $Z = -2.98$, $p = 0.003$, Fig. 1F).”*

P. 26, ll. 12-31: *“We then tested if the individual beta coefficients for ΔES mediated the negative association between mean confidence and OCI-R scores. In other words, we take OCI-R as the independent variable, confidence as the outcome variable, and individual beta coefficients for ΔES as the mediating variable. The c path represents the total effect of OCI-R on confidence (without controlling for the mediating variable); c' , the direct effect of OCI-R on confidence (controlling for the individual beta coefficients for ΔES); b , the effect of the individual beta coefficients for ΔES on confidence, controlling for OCI-R; a , the effect of OCI-R on the individual beta coefficients for ΔES and the product ab , the indirect effect of OCI-R on confidence through the individual beta coefficients for ΔES. We tested statistical significance using a bootstrap test with 10,000 bootstrap samples (<https://github.com/canlab/MediationToolbox>)^{54,55}.*

The rationale for the particular model we tested (OCI-R → ΔES → confidence) is motivated by evidence from the field of metacognition showing a directional link between evidence strength and confidence (i.e., stronger evidence strength preceding higher confidence ratings, as reviewed recently¹⁰⁰). Likewise, there is also evidence for low confidence in OCD⁵³. With respect to directionality assumptions note that here we measure confidence in the restricted context of this particular task, as opposed to a trait-like measure of confidence. Thus, we ask whether the observed lowered confidence associated with high OCI-R scores is mediated through ΔES, rather than whether confidence in a task would be causal with respect to a stable mental health trait (OCI-R).”

Q2.9 Figure 1C seems to indicate that # draws were also used. This should be said in the text.

R2.9 This was in fact stated in the text as follows:

P. 6, ll. 5-7: *“In keeping with the idea of collapsing decision boundaries or urgency signals^{4,30,44,45}, we also included the draw number as a predictor. Draw number additionally predicted whether participants would make a decision ($\beta = 1.799$, $SE = 0.006$, $p < 0.001$)”*

Q2.10 This figure shows on the vertical axes $p(\text{decision})$. How can a probability be negative? Or larger than 1? You 2This figure shows on the vertical axes $p(\text{decision})$. How can a probability be negative? Or larger than 1? You obably mean to write “logistic regression coefficients” or “beta coefficients”, or something like that.

R2.10. Thanks for pointing this out. We have changed the figures so the axis reads “ $p(\text{decide})$ beta coefficients” to avoid any confusion.

Q2.11 “Next, we investigated which of the identified cognitive contributors explained differences in task behaviour in high-OC participants.“ It sounds like you are running first one model, then another, and then another. But in figure 1C you seem to show beta coefficients for all variables you considered. The way this should be done is to decide ahead of time on all possible factors and run a single model. Otherwise you get into horrible multiple comparison problems. What did you actually do when reporting all these p-values up to this point? Single model running one variable at a time?

R2.11 Thanks for raising this. To clarify, in fact we report results from the same model throughout. We introduced the different sets of coefficients sequentially in the text, as we assumed this would be easier to understand for the reader - we first explain findings related to general cognitive factors and then those related to individual differences in a more understandable manner. Acknowledging the confusion this caused for the reviewer, we have revised the text accordingly.

P. 5, ll. 10-16: *“First, to better understand the cognitive contributors to participants’ information gathering behaviour, we predicted on each draw whether a participant would continue sampling information or make a decision ($p(\text{decide})$) using a series of cognitive and individual differences predictors (see Methods for full model). In this task, information accumulates non-monotonically, allowing us to distinguish the contribution of different factors (e.g., current information, total evidence, number of draws) to commitment to a decision.”*

Q2.12 “We found that OC symptoms were linked to reduced weighting of previous evidence $|ES_{d-1}|$ ($= -0.053$, $SE = 0.007$, $p < 0.001$) and particularly of ES ($= -0.085$, $SE = 0.009$, $p < 0.001$).” You probably mean the interaction terms shown in figure 3C? If so, say so.

R2.12 Thanks, this is indeed what we mean – we have written this more explicitly now for these, and the following, interaction results as $ES_{d-1} \times OC$: $\beta = -0.053$, $SE = 0.007$, $p < 0.001$, etc.

Q2.13 “Further, by fitting an additional GLM predicting $p(\text{decide})$ from the current evidence at draw d , $d-1$ and $d-2$ relative to the chosen option per participant, we found that a difference in beta weights for the evidence at draw d and the evidence at draw $d-1$ correlated negatively with the OCI-R score ($s = -0.068$, $p < 0.001$), supporting the notion of a decreased recency effect.” At this point one really worries about multiple comparisons. Testing all these many modes is really not statistically sound. And by the way, the negative beta value says the same as the beta value for delta evidence. I suggest you simply omit this extra analysis.

R2.13 Thank you for this point., However we respectfully disagree. Firstly, our main analysis is testing different hypothesized cognitive (and mental health) contributors to information gathering, all in the same model; thus, multiple comparisons is not an issue. Moreover, this supplementary analysis provides additional information that justifies the separation of evidence contributors. It is thus a relevant analysis that further supports our main findings. Indeed, we included it as a targeted test of a recency effect based on the suggestions from other researchers in the field. The fact that the results converge (as negative beta values) strengthens the credibility of our findings and certainly does not undermine them (as hinted at by this reviewer).

We have clarified this as follows:

P. 6, ll. 31-32 - p. 7, l. 1: *“To follow up on the recency effect suggested by these results, we targeted this by fitting a separate GLMM predicting $p(\text{decide})$ from the current evidence (here, relative to the chosen option) at draw d , $d-1$ and $d-2$ per participant (see Methods).”*

P. 25, ll. 13-27: *“In a follow-up analysis, we tested the recency effect suggested by these results in a more targeted manner, by fitting a GLMM predicting $p(\text{decide})$ from the current evidence at draw d , $d-1$ and $d-2$*

$$p(\text{decide}) \sim ES_{d,\text{total}} + ES_{d-1,\text{total}} + ES_{d-2,\text{total}} \\ + (1 + ES_{d,\text{total}} + ES_{d-1,\text{total}} + ES_{d-2,\text{total}} | \text{participant})$$

with a logit link function, whereby the evidence $ES_{d,\text{total}}$ is defined relative to the chosen option as opposed to the current majority:

$$ES_{d,\text{total}} = ES_{d,\text{chosen}} - ES_{d,\text{unchosen}} \quad (5)$$

To corroborate the individual differences, we found in the recency effect associated with OC symptoms, we derived a metric of recency bias for each participant by fitting a separate generalized linear model (GLM) to each participant analogous to the GLMM

$$p(\text{decide}) \sim ES_{d,\text{total}} + ES_{d-1,\text{total}} + ES_{d-2,\text{total}}$$

with a logit link function and taking the relative difference between the beta coefficients for $ES_{d,\text{total}}$ and $ES_{d-1,\text{total}}$. We then correlated this index of recency bias with the OCI-R score.”

Q2.14 Figure 4A; “Individual differences in OC factor predict significantly decreased decodability within 420-470 ms and 530-560 ms post stimulus (right y-axis). “ I do not understand what the orange curve is, nor what this caption text says.

R2.14 Thanks for pointing out this lack of clarity. We now provide a more detailed explanation for the plot in in the figure caption:

Figure 4: *“Reduced decodability of Δ ES from MEG activity linked to OC symptoms. A Δ ES can be decoded from MEG activity using an iterative multivariate lasso regression (left y-axis: decodability, measured as the correlation between trained and tested data at each timepoint in the epoch, plotted in blue). Individual differences in OC factor predict significantly decreased decodability within 420-470 ms and 530-560 ms post stimulus (right y-axis: beta coefficient for the OC factor predicting the decodability of Δ ES at each timepoint in the epoch from the three factor scores in a GLM per time point, plotted in orange). Blue shaded area represents the standard error.”*

Q2.15 “We found that ES representations were attenuated in high-OC participants, whereby two time periods reached significance using cluster-based permutation testing, namely 420-470 ms and 530-560 ms after stimulus onset (Fig. 4A-B; controlling for other psychiatric factors).” I do not know what this means or how it was done.

R2.15 Thanks for pointing out a lack of clarity. We e have added the following:

P. 13 ll. 24-29: *“On this basis, we assessed how transdiagnostic OC symptoms relate to the decodability of Δ ES by predicting decodability at each time point from the three factor scores in individual GLMs. Using cluster-based permutation testing, we found that Δ ES representations were attenuated in high-OC participants, whereby two time periods reached significance, namely 420-470 ms and 530-560 ms post stimulus onset (Fig. 4A-B, $p < 0.025$ for a two-sided test, cluster-based permutation tests at $t > 2$).”*

In addition, we now include an explanation of the rationale for cluster-based permutation testing, a well-established and commonly used approach, in the Methods section:

P. 28, ll. 10-20: *“Non-parametric cluster-based permutation testing (CBPT¹⁰⁵) was used to assess significant differences in decodability time-courses. CBPT is commonly used in M/EEG analysis to test for significant differences in a given timeseries while accounting for multiple comparisons (e.g., in previous work^{106,107}). Briefly, CBPT takes the temporal structure of the data into account and uses permutation tests (here, $N = 1,000$) to build a null distribution of clusters above a given size (here, threshold $t = 3$) within the same temporal structure, enabling the evaluation of whether an effect of interest exceeds the expected cluster size under the null.”*

Individual differences in decodability were assessed by predicting the decodability at each time point from the three factor scores in individual GLMs. Significant differences in the beta regressor time-courses were assessed by non-parametric CBPT (here, threshold $t = 2$)."

Q2.16 "Neither the decodability of the variable nor the individual differences appear driven by the effect of a commitment to a final decision, as both decodability and individual differences associated with the OC factor remained consistent after omitting the sample immediately preceding the response" I do not understand the rationale of this. What does the previous sample have to do with "commitment" ?

R2.16 We appreciate that our previous wording was unclear. This analysis tested whether the effects reported before were primarily driven by choice trials (where $p(\text{decide})$ is usually the largest) or whether omitting these trials would still yield the same effects. Our results show that the effects hold even in the absence of these trials, which means that they are unlikely to be driven by mere choice-related components (e.g., motor preparations).

We have now reworded the main text accordingly:

P. 13, ll. 29-30 - p. 14, ll. 1-4: *"Neither decodability of the variable nor individual differences in its decodability appear to be driven exclusively by processes related to the draw wherein participants actually declare, as opposed to the longer term evidence accumulation process, as both decodability and individual differences associated with the OC factor remained consistent after omitting the data for the draw immediately preceding the response (see Supplementary Fig. 6A)."*

Q2.17 While Figure 4B seems reasonably clear, the time window seems horribly cherry picked! The methods section on this is not trust inspiring. "Non-parametric cluster-based permutation tests ..." Given the many doubts above, the remaining language in this paragraph is suspect as it is all rather cryptic. This will need to be explained with more clarity.

R2.17 As detailed in the previous responses, cluster-based permutation testing (CBPT) is an established procedure in the M/EEG community - precisely as it addressed multiple comparison problems in data that is assumed to have a spatiotemporal structure (Maris & Oostenveld, 2007). We and others have used this approach across many previous papers (e.g., Hauser et al., 2015; Hunt et al., 2013). As such, the time window is not cherry-picked but instead reflects a statistically significant window using this widely accepted method.

However, the illustrative correlation plot (e.g. 4B) is indeed just a visualisation of this and should not be interpreted as a separate analysis. This is also why we did not specify any correlation coefficient or p-value in that subfigure. Based on this reviewer's feedback, we assume this was not sufficiently clear. We now make this clearer in revised figure captions.

Figure 4: *“Illustration of the attenuated Δ ES decodability within 420-470 ms, plotted here by categorical group, i.e., for participants from the general population with low and high obsessive-compulsive scores (‘low C’ and ‘high C’ respectively), healthy controls (‘Control’), participants with generalized anxiety disorder (‘GAD’), and participants with OCD (‘OCD’). Note this is merely a visualization of the significant difference found in A showing individual data points.”*

Q2.18 “Here, we estimated the lasso regression 2,000 times, using a random subset of 50 sensors on each iteration. The primary benefit of this method is the increased reliability of the individual differences analysis, relative to a single iteration of a multivariate lasso regression using all channels. “ I don’t understand this. Usually one uses cross-validation to test performance. Leaving sensors out seems like a very unusual thing to do? What do you gain from it? Reduce the number of parameters and thus less overstraining? That is what lasso regularization is supposed to do. I have never seen this done, please explain.

R2.18 Thank you for these questions, and we appreciate we were not sufficiently clear in the original main text. The detailed approach was previously implemented within our Centre (Kurth-Nelson et al., 2015), and subsequently used in several other studies (e.g., Liu et al., 2019; Rollwage et al., 2020 - these last additional citations were included in the Methods section). The approach was originally developed to determine the key sensors contributing to the decoding. However, we found the primary advantage in our setting is that it improves the reliability of individual differences results, as is now stated in the methods. This is because decodability curves are always noisy at a single-participant level, such that averaging across many iterations of the lasso improves the stability of the individual differences results by reducing noise of individual decoding traces. Importantly, all the different approaches yielded very similar results, demonstrating our findings are robust and generalize across different methods (e.g., see Supplementary Figures 8 and 9).

We have reworded and expanded the corresponding sections in the methods:

P. 27, ll. 28-31 – p. 28, ll. 1-4: *“Whole-brain MEG activity was used to decode Δ ES using an iterative multivariate lasso regression^{57,103,104}. Here, we estimated the lasso regression 2,000 times, using a random subset of 50 sensors on each iteration. While the method was originally developed to determine which sensors contribute most to the decoding, its primary benefit in this study is that it increases reliability of the individual differences analysis relative to the results from a single iteration of a multivariate lasso regression, by leveraging the many iterations of the lasso (see also Supplementary Materials Alternative analyses of the localisation of the deltaES representation).”*

P. 28, ll. 26-32: *“To investigate which brain areas underpin the decoding, we used a searchlight analysis, such that the lasso regression was estimated 273 times, i.e., for each sensor and its direct neighbors, determined using the Fieldtrip ft_prepare_neighbours function. We chose this method instead of the iterative multivariate lasso regression described above for localization of the effect*

because we and others¹⁰³ have found sensor contribution maps derived from the iterative multivariate lasso regression to be very diffuse (see Supplementary Materials Alternative analyses of the localisation of the deltaES representation)."

Q2.19 "The correlation between the tested and trained data constituted the measure of decodability." What? What do you mean by "correlation" here? Typically you want to report AUC or accuracy or some such thing on test data.

R2.19 Thanks for raising this question. Here, we use correlations because we are predicting a continuous variable as opposed to a categorical one (where AUC is usually used). This approach has been used previously for continuous variables in similar contexts, for example by Eldar et al. (2018) or Toiviainen et al. (2014).

Minor:

Q2.20 Page 5: "general linear mixed models" this is typically called generalized linear mixed effect model.

R2.20 Thanks, this has been corrected.

Q2.21 Figure 2B: there are 5 groups here, but the caption does not define the acronyms or even say what these groups are.

R2.21 Thanks, this has been clarified where appropriate:

Figure 2: *"We used this OC factor to quantify individual differences across all participants, plotted here by categorical group, i.e., by participants from the general population with low and high obsessive-compulsive scores ('low C' and 'high C' respectively), healthy controls ('Control'), participants with generalized anxiety disorder ('GAD'), and participants with OCD ('OCD')."*

Figure 4: *"Illustration of these attenuated Δ ES effects within 420-470 ms, plotted here by categorical group, i.e., for participants from the general population with low and high obsessive-compulsive scores ('low C' and 'high C' respectively), healthy controls ('Control'), participants with generalized anxiety disorder ('GAD'), and participants with OCD ('OCD')."*

Q2.22 All the statistics are missing degrees of freedom or sample number. Please add.

R2.22 We have reviewed the manuscript and are not able to find any instances of this. Kindly point out where there are missing degrees of freedom or sample numbers as we are struggling to

reconcile our reading with the referee's assertion that 'All the statistics are missing degrees of freedom'.

Q2.23 References 51 and 52 are not helpful when pointing to a toolbox. I could not find that M3 toolbox.

R2.23 Apologies for this oversight, we have added the direct link to the Github repository as "<https://github.com/canlab/MediationToolbox>^{54,55}".

Reviewer #3

This study investigates how biases in information gathering contribute to indecisiveness, particularly in individuals with OC traits and OCD. The authors propose a novel information integration bias, where recently acquired information is overweighted during decision-making, operationalized as evidence strength updates (ΔES). Using a large, crowd-sourced dataset ($N = 5,237$), the authors find that attenuated ΔES -weighting is associated with greater indecisiveness across an OC spectrum. This finding is replicated in a lab-based sample ($N = 105$), which includes individuals with OCD and GAD. To explore the neural basis of these biases, the authors use MEG and identify a late neural signal (~ 920 ms post-stimulus) in mediofrontal areas that tracks ΔES . Importantly, high-OC participants show reduced ΔES representation in these regions, while other decision-related processes remain intact.

Strengths:

- Large, well-powered sample provides strong statistical support
- Replication in a controlled clinical population strengthens reliability
- Innovative MEG approach identified neural correlates

Major Comments

Q3.1 The study uses a large-scale, crowd-sourced sample, but only a small sample reports an OCD diagnosis. How do the findings in self-reported high-OC participants compare to clinically diagnosed OCD patients? If available, a separate analysis of diagnosed OCD participants vs. high-OC but non-diagnosed individuals would strengthen clinical relevance.

R3.1 Thank you for raising this. In fact, we conducted this analysis and included it in the supplementary materials under “Self-reported OCD diagnosis effects on information gathering in smartphone population sample”. Briefly, we replicate the key findings, though less strongly:

“We replicated the effect of increased number of draws to a decision in those who self-reported having a current diagnosis of OCD ($N = 129$) vs no lifetime psychiatric disorder ($N = 3,948$), albeit less robustly ($t(4075) = 1.655$, $p = 0.049$, one-sided test). Likewise, the attenuation of the ΔES weight was not as robust and the attenuation in weight of previous evidence ES_{d-1} was not significant for those with self-reported OCD ($OCD \times ES_{d-1}$: $\beta = -0.058$, $SE = 0.043$, $p = 0.093$; $OCD \times \Delta ES$: $\beta = -0.117$, $SE = 0.060$, $p = 0.026$, one-sided tests).”

Q3.2. The decision-making task is highly controlled and involves artificial stimuli (hidden locations and stimuli counts). How well does this task generalize to real-world decision-making scenarios relevant to OCD, such as compulsive checking? Consider discussing ecological validity and whether the observed findings translate to clinical OCD behaviors.

R3.2 Thank you for this question – indeed we have found in the past that the number of draws correlates with the indecisiveness subscale of the Y-BOCS (Hauser et al., 2017). Similarly, excessive information gathering has been found in OCD in a real-life context (Loosen et al., 2021). As the ecological validity is an interesting topic, we now add a section on this in the revised discussion section:

P. 18, ll. 30-32 - p. 19, ll. 1-11: *“Indecisiveness manifests in different ways and with different degrees of severity. In this context, it will be important to establish how well our findings translate to real-life settings. While we have shown in previous work that increased real-life information gathering is associated with OC symptoms²³ and in the current study show that our information gathering paradigm manifests robust and replicable effects linked to OC symptoms, the ecological validity of the task is less evident. We acknowledge this is a highly abstract task designed to target fundamental information gathering mechanisms as well as comply with neuroimaging requirements. That said, we have previously found that information gathering in a version of this task without any sampling constraints is linked to indecisiveness (as rated in the clinician-rated Y-BOCS⁵²). On the opposite end of the spectrum, JTCs in a similar paradigm are consistently linked to clinically reported delusions, indicating good ecological validity for this symptom. However, a more targeted ecological validation of the paradigm could help refine our understanding of the throughline from our findings to putative differences in real-life information gathering in OCD.”*

Q3.3. The study suggests that attenuated Δ ES processing originates in mediofrontal areas, but the functional role of these areas in OC-related decision-making is not fully explored. The mediofrontal cortex is implicated in error monitoring and cognitive control—could altered conflict monitoring rather than evidence weighting per se explain these results? A comparison with previous neuroimaging findings in OCD would help clarify the functional implications of this attenuation. (e.g., Perera, M. P. N., Mallawaarachchi, S., Bailey, N. W., Murphy, O. W., & Fitzgerald, P. B. (2023). Obsessive–compulsive disorder (OCD) is associated with increased engagement of frontal brain regions across multiple event-related potentials. *Psychological Medicine*, 53(15), 7287-7299.)

R3.3 Thanks for this interesting comment. We now include a reference to the substantial body of literature that implicates fronto-striatal circuitry in OCD, including the paper cited here.

P. 17, ll. 6-9: *“More broadly, the localisation of Δ ES decodability effect accords with accumulating evidence for alterations in fronto-striatal circuitry in OCD, whereby these findings are often related to heightened error monitoring and reward prediction error response⁶⁶⁻⁶⁸.”*

Q3.4 The study demonstrates neural alterations in information processing in high-OC individuals, but it does not discuss whether these findings translate to clinically diagnosed OCD patients.

R3.4 Thanks for raising this issue. We indeed adopt a transdiagnostic approach, where OC symptoms are not restricted to those with a formal OCD diagnosis. Such an approach is very much in keeping with previous studies which show that neurocognitive mechanisms seem more closely tied to a transdiagnostic construct, rather than putative categorical diagnostic entities (Gillan et al., 2020). Based on this reviewer's suggestion, we conducted a subgroup analysis, comparing Δ ES in those who had a formal OCD diagnosis (N = 29) with a matched control group (N = 19). While we find similar clusters profiles to those reported in the main manuscript, these did not reach significance likely due to the smaller sample size. We now address this in the discussion.

P. 18, ll. 11-26: *"In this study, we focus on transdiagnostic OC effects within the general population as well as across different clinical diagnoses. While many of our (primarily behavioural) effects hold in more traditional patient vs controls sub-samples, these tended to be less strong (e.g., MEG differences did not reach significance, data not shown) – possibly due to a much smaller sample size. [...] Nevertheless, it is yet to be determined whether these information gathering biases are best described using transdiagnostic approaches, as adopted in previous studies where dimensional measures outperformed traditional diagnostic categorical comparisons⁵⁶. Specifically, it remains to be established, first, whether our findings hold in traditional OCD patient case-control studies, and second, to what extent these hold across diagnoses."*

Q3.5. Could these results inform biomarker development for OCD or potential neuromodulation targets (e.g., tACS or TMS over mediofrontal areas)? Adding a discussion on how these findings may be applied to clinical interventions would enhance the study's translational value.

R3.5 Thank you for this suggestion, we now allude to this in the discussion:

P. 18, ll. 2-4: *"Indeed, the mediofrontal regions which appear most implicated in the attenuation of the Δ ES representation provides a tantalising target for more focused future investigations, as well as for potential therapeutic interventions (e.g., electrical or magnetic stimulation⁷¹."*

Minor Comments

Q3.6 In Pg 18, methods: the authors talk about two studies ('in both studies'), but it is not clear what the two studies exactly are. Please consider re-writing in two sections (e.g., study 1, study 2)

R3.6 We apologize for the lack of clarity, we have rephrased this as "in both online and in-person studies" where applicable.

Q3.7 It would be helpful if a PRISMA chart was included in the paper to show how participant recruitment and exclusion occurred.

R3.7 Thanks for this suggestion. It is our understanding that PRISMA is only applicable for meta-analyses. However, we acknowledge that such a figure would be helpful and now include a figure in the supplementary materials:

Supplementary Figure 2: Participant recruitment and exclusion procedure.

Q3.8 Pg 19, “Group demographics, medication percentages, and comorbidities for the final sample included in the analysis are reported in Supplementary Table 1.” It appears to be in Table 2 within the supplementary material.

R3.8 Thanks for catching this error, it has been corrected.

Q3.9. The discussion section could be improved by adding a “Limitations and Future Directions” sub-section.

R3.9 Thanks for this suggestion. We have made this section more explicit in the revised discussion.

P. 18, ll. 11-32 - p. 19, ll. 1-11: “In this study, we focus on transdiagnostic OC effects within the general population as well as across different clinical diagnoses. While many of our (primarily behavioural) effects hold in more traditional patient vs controls sub-samples, these tended to be less strong (e.g., MEG differences did not reach significance, data not shown) – possibly due to a much smaller sample size. Conversely, we find that OC symptoms are not exclusive to OCD patients but are also prominent in some GAD patients and non-clinical, high OC participants in

our sample (see Fig. 2B). While data on indecisiveness in clinically diagnosed GAD is, to the best of our knowledge, lacking, our findings align with previous reports of high indecisiveness in undiagnosed individuals with probable GAD⁷⁹ as well as in those with high levels of symptoms of anxiety and depression^{80,81}. Thus, we propose that indecisiveness and excessive information gathering may represent transdiagnostic symptoms not limited to OCD.

Nevertheless, it is yet to be determined whether these information gathering biases are consistently best described using transdiagnostic approaches adopted in previous studies where dimensional measures outperformed traditional diagnostic categories⁵⁶. Specifically, it remains to be established, first, whether our findings hold in traditional OCD patient case-control studies, and second, to what extent these hold across diagnoses. Indecisiveness can also be a symptom of both depression and dependent personality disorder¹², yet understanding the overlap in symptom phenomenology and underlying mechanisms will require future studies that include these populations alongside those with OCD and GAD.

Indecisiveness manifests in different ways and with different degrees of severity. In this context, it will be important to establish how well our findings translate to real-life settings. While we have shown in previous work that increased real-life information gathering is associated with OC symptoms²³ and that the current information gathering paradigm shows robust and replicable effects linked to OC symptoms, the ecological validity of this task is less evident. We acknowledge this is a highly abstract task designed to target fundamental information gathering mechanisms as well as comply with neuroimaging requirements. That said, we previously found that information gathering in a version of this task without any sampling constraints is linked to indecisiveness (as rated in the clinician-rated Y-BOCS⁵²). At the opposite end of the spectrum, JTCs in a similar paradigm are consistently linked to clinically reported delusions, indicating good ecological validity for this symptom. However, a more targeted ecological validation of the paradigm could help refine our understanding of the throughline from our findings to differences in real-life information gathering in OCD.”

Reviewer #3 (Remarks to the Author):

The points raised have been addressed sufficiently.

R3.0: We appreciate the reviewer's feedback and are pleased to hear that the revisions have sufficiently addressed the concerns.

Reviewer #4 (Remarks to the Author):

This is an important study that bridges population-level computational psychiatry with in-lab brain imaging. The authors identify attenuated weighting of recent evidence (Δ ES) as a computational mechanism linked to indecisiveness and obsessive-compulsive symptoms. The combination of very large online samples, replication in a clinical cohort, and convergent MEG data is rare in the field and gives the findings unusual strength.

The evidential base is particularly strong. More than 5,000 online participants provide statistical power and generalizability, and the key findings are reproduced in a clinical sample including OCD and GAD patients. The MEG analyses then reveal a temporally distinct neural signal of Δ ES in mediofrontal regions, attenuated in high-OC individuals. Taken together, the convergence across behavioral, clinical, and neural data sets a high standard for explanatory strength.

Concerns raised during the first round of review have been addressed thoroughly. Questions about statistical robustness are resolved by the addition of preregistration details, full model comparison tables, cross-validation, and parameter recovery analyses. These show that the Δ ES parameter is stable and not the result of overfitting or selective reporting. Likewise, concerns about the neural evidence have been met with convergent source reconstructions and more cautious framing. The late Δ ES signal, appearing around 920 ms and present even on correct trials, is distinct from error-monitoring. Although MEG localization inevitably carries uncertainty, the convergence of methods and the circumspect interpretation are sufficient to support the authors' claims.

The authors themselves note that their paradigm is optimized to isolate evidence accumulation and does not capture all forms of indecision seen in daily life. I agree this is an important limitation and would suggest foregrounding it slightly more in the discussion, so that readers appreciate both the strength of the mechanistic isolation and the limits of ecological generalization. Presentation could also be improved by clearer cross-referencing between main figures and supplementary analyses, which would help readers navigate the technical results.

Overall, this is an ambitious and carefully executed piece of work. The revision has strengthened the manuscript considerably, and in my judgment all substantive concerns have now been resolved. I see no reason to delay publication further and recommend acceptance after only minor editorial adjustment.

R3.0: We appreciate the reviewer's positive assessment of our manuscript with regard to the strengths of the work as well as the comprehensive revision. In line with your suggestion, we have reworded the paragraph discussing limitations to more clearly foreground the trade-off between mechanism isolation and ecological generalizability (see below). We have also reviewed the manuscript to ensure consistent and clear cross-referencing between main figures and supplementary analyses to aid navigation of the technical results.

P. 19, ll. 1-13: *“Further, indecisiveness manifests in different ways and with different degrees of severity, and our highly abstract task does not capture this complexity. The strength of this abstraction is that it enables the isolation of fundamental evidence accumulation processes and meets neuroimaging constraints, yet it comes at the cost of ecological generalizability. While previous work has linked increased real-life information gathering to OC symptoms, and the current paradigm shows robust and replicable effects in relation to OC symptoms, it will be important to establish how well our findings translate to real-life settings.”*